# Genomic Selection for Early Growth Traits in Inner Mongolian Cashmere Goats Using ABLUP, GBLUP, and ssGBLUP Methods

**DOI:** 10.3390/ani15121733

**Published:** 2025-06-12

**Authors:** Tao Zhang, Linyu Gao, Bohan Zhou, Qi Xu, Yifan Liu, Jinquan Li, Qi Lv, Yanjun Zhang, Ruijun Wang, Rui Su, Zhiying Wang

**Affiliations:** 1College of Animal Science, Inner Mongolia Agricultural University, Hohhot 010018, China; 2Grassland Research Institute, Chinese Academy of Agricultural Sciences, Hohhot 010018, China; 3Inner Mongolia Key Laboratory of Sheep & Goat Genetics Breeding and Reproduction, Hohhot 010018, China; 4Key Laboratory of Mutton Sheep & Goat Genetics and Breeding, Ministry of Agriculture and Rural Affairs, Hohhot 010018, China

**Keywords:** genomic selection, accuracy, GEBV, Inner Mongolian cashmere goats

## Abstract

This study focused on improving the growth rates of Inner Mongolian cashmere goats (IMCGs) by identifying the most effective genomic selection methods for early growth traits. Faster growth rates are important for farmers because they can lead to increased meat production, ultimately enhancing the economic value of these goats. We analyzed data from 2256 cashmere goats, looking at traits such as birth weight, weaning weight, daily weight gain, and yearling weight. By using advanced statistical models, we aimed to determine which method would provide the most accurate predictions for breeding values based on genetic information. Our research found that specific factors like birth year, herd, sex, birth type, and the age of dams play important roles in growth traits. Among the models we tested, the one that included additive genetic effects from both the individual and its mother, as well as environmental effects, was best. Notably, the method known as ssGBLUP offered the highest accuracy in predicting breeding values, making it the preferred choice for breeding programs aimed at enhancing growth rates in IMCGs. In conclusion, we recommend using the ssGBLUP method for improving breeding efficiency in the growth traits of IMCGs.

## 1. Introduction

The Inner Mongolian cashmere goat (IMCG), a highly regarded breed known worldwide, has been developed over a long period through natural selection and artificial breeding for both cashmere and meat production [1]. This breed has the advantages of a high cashmere production, good cashmere quality, and stable genetic architecture. It is used as the paternal parent for the cultivation of many cashmere goat breeds in China and was listed as one of the first breeds for the protection of genetic resources of livestock and poultry in China.

Early growth traits have strong implications for the reproductive and production performance of animals [2,3,4]. The growth and development of young lambs are directly linked to the economic benefits of livestock production, serving as key indicators of growth rate, health status, and overall productivity. Gätjens [5] reported that individuals born small had a higher risk for cardio-metabolic disease. Early growth traits, such as birth weight and weaning weight, are the basis for selection in genetic improvement programs for meat production due to their strong association with each other [6]. Therefore, the assessment of individual growth performance is imperative for determining the genetic potential of breeds and subsequently devising effective genetic improvement programs.

Genomic selection was described by Meuwissen et al. [7]. It is widely used in the selection of superior animals and plants [8,9,10]. There have been numerous reports on the genetic evaluation of early growth traits in livestock. Lavvaf and Noshary [11] used a single-trait animal model to perform a genetic evaluation of early growth traits in the Lori breed of sheep. Early growth traits were significantly affected by direct additive genetic effects, direct maternal genetic effects, and environmental effects. Balasundaram et al. [12] evaluated the genetic parameters of the growth traits and quantitative genetic metrics of Mecheri sheep in Tamil Nadu. The best model included direct and maternal genetic effects as random effects with no covariance.

Due to their economic importance, early growth traits in most species have been subjected to genomic selection to improve breeding efficiency. Fei Ge et al. [13] evaluated the accuracy of genomic prediction for growth traits at weaning and yearling ages in yak. The result indicated that the prediction accuracy for early growth traits in yak ranged from 0.147 to 0.391. Terakado et al. [14] compared methods for predicting genomic breeding values for growth traits in Nellore cattle. Bayesian methods provided slightly more accurate predictions of genomic breeding values than the GBLUP. Song et al. [15] performed genomic prediction for growth traits in pig using an admixed reference population. It was demonstrated that ssGBLUP was the best method for genomic selection in pigs. Tianfei Liu et al. [16] evaluated the accuracy of genomic prediction for growth traits in Chinese triple-yellow chickens. The accuracy of the genomic prediction of growth traits ranged from 0.448 to 0.468. However, there are few reports on the genomic selection of early growth traits in goats. The purpose of this study is to determine the best model and method for the genomic prediction of early growth traits in IMCGs. This is of great significance for performing the genome selection of early growth traits in Inner Mongolian cashmere goats.

## 2. Materials and Methods

### 2.1. Data and Population Sources

The data were collected from the Inner Mongolia Yi-Wei White Cashmere Goat Limited Liability Company, located in the southwestern part of Inner Mongolia (Ordos, Inner Mongolia, China). Birth weight (BW), weaning weight (WW), daily weight gain before weaning (DWG), and yearling weight (YW) were considered in this study. The phenotypes of the above traits were recorded from 2014 to 2019. The total number of individuals with pedigree records was 14,165 from 1990 to 2019, from which 2256 were genotyped using the 70K SNP chip (Illumina Inc., San Diego, CA, USA).

### 2.2. Data Preprocessing

For the phenotype data of the early growth traits collected from this population, an Excel spreadsheet was used for preprocessing. Firstly, the individuals with recorded genotypes were extracted from a database of breeding information, and the outliers for each trait were further excluded. Values outside the range of the mean plus or minus 2.58 times the standard deviation were defined as outliers. Then, quality control of the genotype data was performed using PLINKv1.90 software [17]. The quality control criteria were as follows: (1) exclude individuals with SNP missing rate of >10%; (2) exclude SNPs with individual missing rate of >10%; (3) exclude SNPs with minor allele frequency (MAF) of <5%; and (4) exclude SNPs violating the Hardy–Weinberg equilibrium (HWE) at *p* < 1 × 10^−5^. The missing genotypes were imputed using Beagle [18]. Finally, a total of 50,728 SNPs in 2256 individuals remained for subsequent analyses.

### 2.3. Determination of Fixed and Random Effects

The determination of fixed and random effects is essential for genetic evaluation. The fixed effects considered in this study include sex (male or female), year of birth (2014–2021), herds (1–12, including 6 adult dam herds, 3 yearling dam herds, 2 yearling ram herds, and 1 adult ram herd), age of dam (2–8 years old), and birth type (single, twins, and triplets). The environmental factors affecting early growth traits in IMCGs were further determined by constructing generalized linear models of the phenotype and environmental effects using the GLM procedure of SAS 9.2 [19]. The generalized linear model is as follows.y_ijklmn_ = u + S_i_ + Y_j_ + H_k_ + D_l_ + B_m_ + e_ijklmn,_
where y is the observations of individuals in each trait, u is the overall mean, S_i_ is the effect of the ith sex, Y_j_ is the effect of the jth year of birth, H_k_ is the effect of the kth herds, D_l_ is the effect of the lth dam’s age, B_m_ is the effect of the mth birth type, and e_ijklmn_ is the random error.

After determining the fixed effects, four single-trait animal models with various combinations of individual and maternal effects were constructed. The formula of each model is as follows:
$$\begin{array}{c} \mathrm{Model}\;1:\;y=\mathrm{Xb}+Z_{1}\;a+e;\\ \mathrm{Model}\;2:\;y=\mathrm{Xb}+Z_{1}\;a+Z_{2}\;m+e;\\ \mathrm{Model}\;3:\;y=\mathrm{Xb}+Z_{1}\;a+Z_{2}\;m+Z_{3}\;c+e\;\mathrm{cov}(a,m)=0;\\ \mathrm{Model}\;4:\;y=\mathrm{Xb}+Z_{1}\;a+Z_{2}\;m+Z_{3}\;c+e\;\mathrm{cov}(a,m)\neq 0; \end{array}$$
where y is the vector of observations for each trait, b is the vector of fixed effects, a is the vector of individual additive genetic effects, m is the vector of maternal genetic effects, c is the vector of maternal permanent environmental effects, and X, Z_1_, Z_2_, and Z_3_ are the incidence matrices for the effects in b, a, m, and c, respectively. e is the vector of residual effects. The difference between Model 3 and Model 4 is the presence or absence of covariance between the direct additive and maternal genetic effects. The total heritability was calculated by the following formula.hT2=(σa2+0.5σm2+1.5σam)σp2,
where $h_{T}^{2}$ is the total heritability. $\sigma_{a}^{2}$ is the individual additive genetic variance. $\sigma_{m}^{2}$ is the maternal genetic variance. $\sigma_{am}$ is the covariance between the individual direct additive genetic effect and maternal genetic effect.

The most appropriate model was obtained using the likelihood ratio test [20]. The formula used to calculate the likelihood ratio value was as follows:$$LR=-2Log\left(\frac{L_{1}}{L_{2}}\right),$$
where LR is the value of the likelihood ratio, and LogL_1_ and LogL_2_ are the maximum likelihood function values for Model 1 and Model 2, respectively. Model 1 was a submodel of Model 2 [21]. The LR statistic follows a chi-square distribution (ℵ^2^), with degrees of freedom equal to the difference in the number of parameters between the two models. The best model can be determined by comparing the LR value to the critical value of the chi-square distribution.

### 2.4. Estimates of Variance Components and Genetic Parameters

The variance components and genetic parameters were estimated for each early growth trait with three methods (ABLUP, GBLUP, and ssGBLUP) using the ASRgenomics package of ASREML4.2 software [22]. For each method, the mixed animal model equations were as follows.

ABLUP: The A matrix was constructed using the pedigree.$$\left[\begin{array}{cc} X^{\prime }Z & X^{\prime }Z\\ X^{\prime }Z & Z^{\prime }Z+{\frac{\sigma_{e}^{2}}{\sigma_{a}^{2}}A}^{-1} \end{array}\right]\left[\begin{array}{c} \hat{b}\\ \hat{a} \end{array}\right]=\left[\begin{array}{c} X^{\prime }y\\ Z^{\prime }y \end{array}\right],$$
where $X^{\prime }$ is the transpose matrix of the incidence matrix of fixed effects. Z is the incidence matrix for the effects in individual additive genetic effects. $Z^{\prime }$ is the transpose matrix of Z. $\sigma_{a}^{2}$ is the variance of the individual additive genetic effects. $\sigma_{e}^{2}$ is the residual variance. A^−1^ is the inverse matrix of A.

GBLUP: The G matrix was constructed using genotype data.$$G=\frac{ZZ^{\prime }}{2\sum \;P_{i}\;(1-P_{i})}$$

In the G matrix above, $P_{i}\;$ refers to the allele frequency of the jth marker.$$\left[\begin{array}{cc} X^{\prime }X & X^{\prime }Z\\ Z^{\prime }X & Z^{\prime }Z+G^{-1}\frac{\sigma_{e}^{2}}{\sigma_{a}^{2}} \end{array}\right]\left[\begin{array}{c} b\\ \mu \end{array}\right]=\left[\begin{array}{c} X^{\prime }y\\ Z^{\prime }y \end{array}\right],$$
where $G^{-1}$ is the inverse matrix of G.

ssGBLUP: The H matrix was constructed by combining the pedigree and genotype.$$H=\left[\begin{array}{cc} H_{11} & H_{12}\\ H_{21} & H_{22} \end{array}\right],\;=\left[\begin{array}{cc} A_{11}+A_{12}A_{22}^{-1}\left(G-A_{22}\right)A_{22}^{-1}A_{21} & GA_{12}A_{22}^{-1}\\ GA_{22}^{-1}A_{21} & G \end{array}\right],$$$$H^{-1}=A^{-1}+\left[\begin{array}{cc} 0 & 0\\ 0 & G^{-1}-A_{22}^{-1} \end{array}\right],$$
where $A_{11}$ is the relationship matrix among known pedigree individuals, $A_{12}$ and $A_{21}$ are the relationship matrices between known pedigree individuals and individuals with molecular marker data, respectively, and $A_{22}$ is the pedigree-based relationship matrix among individuals with molecular marker data. G is the genomic relationship matrix among individuals with molecular marker data. $A^{-1}\;$ is the inverse of the submatrix $A_{11}$, and $A_{22}^{-1}\;$ is the inverse of the submatrix $A_{22}$. The meanings of the other letters are the same as in the ABLUP mixed model equations.

### 2.5. Assessment of the Genomic Prediction Accuracy

The research population was divided into five groups, four of which were combined as the reference group and the remaining as the validation group. Each group was sequentially used as the validation group, and this was repeated five times; finally, the average of the correlation coefficients between the corrected phenotype values and the breeding values obtained from the five repetitions was used as a criterion for evaluating the genomic prediction accuracy. When the result is closer to 1, its prediction ability is better [23]. The formula was as follows:$$r=\frac{cov(a,p)}{\sqrt{var\left(a\right)var(p)}},$$
where cov(a, p) is the covariance of the estimated breeding and phenotype values in each trait, respectively. var(a) and var(p) are the variances of the estimated breeding values and phenotype values in each trait, respectively.

The formula for the reliability of the genomic estimated breeding value (GEBV) is as follows:$$r_{i}^{2}=1-\frac{{PEV}_{i}}{\sigma_{a}^{2}},$$
where ${PEV}_{i}$ is the prediction error variance of the ith individual, which was obtained by inverting the mixed model equations. It reflects the uncertainty of the estimated breeding value. $\sigma_{a}^{2}$ is the additive genetic variance.

## 3. Results

### 3.1. Descriptive Statistics of Early Growth Traits in IMCGs

The basic descriptive statistics of the phenotype data of early growth traits in Inner Mongolian cashmere goats are shown in Table 1. The mean values of BW, WW, DWG, and YW were 2.76 kg, 21.97 kg, 0.17 kg, and 37.20 kg, respectively, and the coefficients of variation ranged from 15.42% to 21.04%. It was found that the differences in each trait among individuals were small.

### 3.2. Determination of Fixed Effects

The analysis of variance of non-genetic factors, including sex, year of birth, herd, age of dam, and birth type, on the early growth traits of IMCGs is shown in Table 2. It is clear that year of birth and herds had a significant effect on BW (*p* < 0.05). The effect of sex on the four traits was highly significant (*p* < 0.001). Year of birth had a highly significant effect on DWG and WW (*p* < 0.001). Herd had a highly significant impact on WW and DWG (*p* < 0.01) and on YW (*p* < 0.001). The age of the dam had a highly significant effect on all the traits in this study (*p* < 0.01). The impact of birth type was highly significant for BW, WW, and DWG (*p* < 0.001) and YW (*p* < 0.01). Therefore, all the factors were included as fixed effects in the animal models used for genetic evaluation in this study.

### 3.3. Model Comparison

The likelihood ratio test values of each model for the early growth traits of IMCGs are shown in Table 3. The value of −2lnL for the early growth traits of IMCGs in Model 4 under each method was lower than that in the other models, indicating that Model 4 is the best model for estimating the genomic breeding values of the early growth traits of IMCGs. In addition, the chi-square test was performed to evaluate the reliability of the models. The results are shown in Table 4. For each trait with three methods, significant or highly significant differences between Model 4 and the other models were observed. No significant differences were observed between Model 1 and Model 2 for YW using either the ABLUP or GBLUP methods. Similarly, Model 1 and Model 3 showed no significant differences for YW under the ABLUP method. For BW and YW, comparisons between Model 2 and Model 3 using the ABLUP method also revealed no significant differences. Additionally, Model 2 and Model 3 did not differ significantly for DWG when analyzed with the GBLUP method.

### 3.4. Estimation of Variance Components and Genetic Parameters

It was confirmed that Model 4 is the best model for the genetic evaluation of early growth traits in IMCGs. The variance components and genetic parameters obtained by the ABLUP, GBLUP, and ssGBLUP methods for each trait in this study under the optimal model are shown in Table 5, Table 6, Table 7 and Table 8. For BW, the range of direct additive heritability with the three methods was 0.09–0.11, the maternal additive heritability was 0.06–0.19, and the contribution of the maternal permanent environmental effect to the phenotypic variance was 2.45 × 10^−8^–0.12. The covariance between the direct additive genetic effects and maternal additive genetic effects was negative in the ABLUP and GBLUP analyses. The correlation coefficients were −0.41 and −0.35, respectively (Table 5). For WW, the range of direct additive heritability with the three methods was 0.17–0.43, the maternal additive heritability was 0.14–0.22, and the contribution of the maternal permanent environmental effect to the phenotypic variance was 1.90 × 10^−8^ to 1.36 × 10^−5^. The covariance between the direct additive genetic effects and the maternal additive genetic effects was negative. The correlation coefficients varied from −0.54 to −0.91 (Table 6). For DWG, the range of direct additive heritability with the three methods was 0.12–0.15, the maternal additive heritability was 0.08–0.11, and the contribution of the maternal permanent environmental effect to the phenotypic variance was 5.84 × 10^−3^ to 7.49 × 10^−3^. The covariance between the direct and maternal additive genetic effects was negative. The correlation coefficients ranged from −0.41 to −0.32 (Table 7). For YW, the range of direct additive heritability with the three methods was 0.23–0.32, the maternal additive heritability was 0.12–0.26, and the contribution of the maternal permanent environmental effect to the phenotypic variance was 1.6 × 10^−5^ to 2.0 × 10^−5^. The covariance between the direct additive and maternal additive genetic effects was negative. The correlation coefficients ranged from −0.84 to −0.56 (Table 8).

Overall, the contribution of the maternal additive genetic effects to the phenotype was relatively large, while the contribution of the maternal permanent environmental effects to the phenotype was relatively small. The heritability of the weaning weight and yearling weight of IMCGs was medium to high, while the heritability of the other two traits was moderate to low.

### 3.5. Evaluation of the Accuracy and Reliability of the Estimated Genomic Breeding Values

The accuracy and reliability of the genomic estimated breeding values (GEBVs) with the three methods for early growth traits in IMCGs are shown in Table 9. The accuracy of the GEBV with the three methods for BW, WW, DWG, and YW was 0.52–0.61, 0.68–0.70, 0.59–0.70, and 0.60–0.67, respectively. It was found that the accuracy and reliability of each trait with the ssGBLUP method were the highest, which were significantly higher than with the other two methods. The lowest accuracy of the GEBV was observed with the ABLUP methods.

## 4. Discussion

Early growth traits are not only an important index for assessing the growth and development of livestock, but also play an important role in a series of production and economic benefits of livestock. There are many factors affecting the early growth traits of cashmere goats, such as sex, birth type, age of the dam, and herd. In addition, other environmental factors such as nutrition and disease also have some effects on the early growth traits of livestock [24]. In this study, all factors, including sex, age of the dam, birth type, herd, and years of measurement, had significant effects on the early traits of IMCGs. Wang et al. [25] showed that year of birth, herd, birth type, sex, and mother’s age had an effect on the preweaning traits of Inner Mongolian Arbas cashmere goats, which is in agreement with the results of this study. Mahala [26] indicated that sex had significant effect on the early growth traits of Avikalin sheep. Lalit et al. [27] showed that year of birth and sex had significant effects on the birth weight, weaning weight, daily weight gain, and weekly weight of Harnali sheep. Jalil-Sarghale [28] showed that year of birth, sex, and birth type had significant effects on birth weight, weaning weight, and weekly weight, which is consistent with the results of our study. Mohammadi et al. [29] reported that year of birth, birth type, sex, and age of the dam had significant effects on the birth and weaning weights of Zandi lambs. Therefore, it can be seen that early growth traits for goats and sheep are greatly influenced by environmental factors. The observed variations could potentially be attributed to interannual differences in temperature, humidity, and pasture conditions, as well as divergent feeding management practices adopted by different herders [30]. In our study, the early growth weights in males were higher than those in females. This may be attributed to the fact that male lambs have a stronger growth intensity than ewes. The influence of age of the dam on early growth traits could be associated with the physiological status of the ewe. Additionally, the competition for nutrients between twin lambs results in a reduced average absorption of nutrients per lamb during the embryonic stage of development, leading to single lambs having higher early body weights compared to twin lambs [31].

As important economic traits, it is necessary to perform accurate genetic assessments of early growth traits. In this study, four animal models were constructed to assess the maternal genetic effect, maternal environmental effect, and covariance between the direct additive effect and maternal genetic effect. The model including all four effects estimated the genetic parameters of the early growth traits in IMCGs best. Illa et al. [32] reported that the direct additive effect, maternal genetic effect, and covariance between the direct additive effect and maternal genetic effect had significant effects on the average daily body weight gain in Nellore sheep. Since the conception rate, lactation ability, and litter protection ability of ewes are affected by both genetics and environment, maternal effects can be divided into maternal genetic effects and maternal environmental effects [33]. Hoque et al. [34] reported that the accuracy of the genetic evaluation of early body weight traits in Japanese Black cattle will decrease if the maternal effects are ignored in the animal model. Gowane et al. [35] and Hanford et al. [36] compared different models for estimating the genetic parameters for the body weight of Bharat Merino sheep and Columbian sheep. The results indicated that maternal additive genetic effects had significant effects on the early growth traits of sheep. Dige et al. [37] reported that the direct additive genetic effect, maternal genetic effects, maternal environmental effects, and covariance between individual additive genetic effects and maternal additive genetic effects had significant effects on growth and feed efficiency traits in Jamunapari goats, which is consistent with the results in our study. However, Ulutas [38] suggested that animal models with only the direct additive genetic effect and maternal genetic effect should be used for the genetic evaluation of preweaning in Suckler cattle. The differences among the results of the above studies may be due to differences in the genetic basis of breeds or population sizes.

In this study, the range of heritability for birth weight was calculated to be 0.09–0.11. Cloete et al. [39] reported that the heritability for birth weight of Merino lambs was 0.16. Di et al. [40] indicated that the heritability for birth weight of ultrafine Chinese Merinos was 0.15. It can be seen that the birth weight of lambs is a trait with a moderate to low heritability. The range of heritability for weaning weight of IMCGs in this study was 0.17–0.43. Menezes et al. [41] demonstrated that the heritability for weaning weight in Boer goats was 0.28, which is higher than that in our study. This may be due to the differences between breeds. The heritability of daily weight gain of IMCGs was consistent with the results in Boer goats from Zhang et al. [42]. The heritability of yearling weight of IMCGs ranged from 0.20 to 0.32, which is similar to the results from Gowane et al. [35] and Abdalla et al. [43]. However, some studies illustrated that the heritability of yearling traits in South African Angora goats and Black Bengal goats was high [44,45]. The differences among the results between studies may be due to various geographic environments, management conditions, and sample size.

As a breeding technology, genomic selection is used to derive genomic breeding values using genetic markers. It has been performed in dairy cattle, beef cattle, pigs, fish, and so on. Genomic selection can effectively improve the genetic progress of livestock and poultry, reduce the cost of progeny testing, and shorten the generation interval, which has been a hot research topic in livestock and poultry breeding in recent years [46]. In this study, three methods, including ABLUP, GBLUP, and ssGBLUP, were used to achieve estimations of the genomic breeding values for early growth traits in IMCGs. It was found that the accuracy and reliability of the GEBV of early growth traits in IMCGs was highest using the ssGBLUP method. Hayes et al. [47] reported that the accuracy of the GEBV of milk production traits with the GBLUP method was almost as high as that with Bayes. Siavash et al. [48] compared the prediction accuracy of the genomic selection of slaughter traits in Canadian regional pig breeds using the BLUP, GBLUP, ssGBLUP, and Bayes methods. The results showed that the ssGBLUP method had a higher prediction accuracy than the other methods. Mancisidor et al. [49] analyzed the accuracy of the genomic selection of important economic traits in Huacaya alpacas with BLUP, ssGBLUP, and other methods. The ssGBLUP method was better for medium- and high-heritability traits when genomic prediction was carried out on small samples. Abdalla et al. [43] reported that the accuracy of the GEBV using ssGBLUP (0.51) was higher than that with ABLUP (0.35). Thus, it can be seen that genomic selection can improve the accuracy of the estimated breeding value. The ssGBLUP method fully utilizes pedigree and genotype information to estimate breeding values, which may be the reason for its high accuracy in genetic evaluation.

## 5. Conclusions

Animal models for the genetic evaluation of early growth traits in IMCGs should include direct additive genetic effects, maternal genetic effects, maternal environmental effects, and the covariance between the direct additive and maternal genetic effects. The maternal effect has an impact on the early growth traits of IMCGs. The direct additive heritability of birth weight and daily weight gain were 0.09–0.11 and 0.12–0.15, respectively. Both were low heritable traits. Therefore, selection for BW and DWG in IMCGs based on the GEBV will result in slow genetic progress. The direct heritability estimates for WW and YW were moderate (0.17–0.43 and 0.20–0.32, respectively). Thus, rapid genetic improvement can be obtained. Furthermore, when appropriate selection intensity, observed phenotypic variation, and potential generation interval reduction through genomic selection are collectively considered, more rapid genetic progress can be anticipated. Additionally, we determined that the accuracy of the GEBV could be improved by performing genomic selection with the ssGBLUP method.

## Figures and Tables

**Table 1 animals-15-01733-t001:** Basic descriptive statistics of early growth traits in IMCGs.

Traits	No.	Mean	C.V (%)	Max	Min
BW (kg)	1811	2.76	16.96	4.53	1.00
WW (kg)	1809	21.97	15.42	34.77	13.60
DWG (kg/d)	967	0.17	17.79	0.27	0.09
YW (kg)	1953	37.20	21.04	80.50	20.50

Note: BW: birth weight, WW: weaning weight, DWG: daily weight gain before weaning, YW: yearling weight, No.: number of individuals, C.V: coefficient of variation, Max: maximum, Min: minimum.

**Table 2 animals-15-01733-t002:** Effects of non-genetic factors on early growth traits in IMCGs.

	Traits	BW		WW		DWG		YW	
Factors		F-Value	*p*-Value	F-Value	*p*-Value	F-Value	*p*-Value	F-Value	*p*-Value
Sex		167.02	$2.0\times 10^{-16}$ ***	1272.77	$2.0\times 10^{-16}$ ***	838.74	$2.0\times 10^{-16}$ ***	436.31	$2.0\times 10^{-16}$ ***
Year of birth		6.18	0.0131 *	6.51	0.0022 **	29.24	$8.1\times 10^{-8}$ ***	255.27	$2.0\times 10^{-16}$ ***
Herds		6.04	0.0142 *	7.21	0.0073 **	9.01	0.0093 **	491.17	$2.0\times 10^{-16}$ ***
Age of dam		10.08	0.0029 **	14.27	0.0002 **	10.48	0.0013 **	7.03	0.0086 **
Birth type		229.33	$2.0\times 10^{-16}$ ***	548.10	$2.0\times 10^{-16}$ ***	195.22	$2.0\times 10^{-16}$ ***	7.52	0.0062 **

Note: BW: birth weight, WW: weaning weight, DWG: daily weight gain before weaning, YW: yearling weight, ***: indicates highly significant difference (*p* < 0.001); **: indicates highly significant difference (*p* < 0.01); *: indicates significant difference (*p* < 0.05).

**Table 3 animals-15-01733-t003:** The values of −2*lnL* in each model for early growth traits with the three methods.

Traits	Model	−2*lnL*		
		ABLUP	GBLUP	ssGBLUP
BW	1	−811.4	−816.1	−810.9
	2	−837.8	−841.4	−815.6
	3	−842.0	−841.4	−843.5
	**4**	**−849.5**	**−847.4**	**−849.9**
WW	1	4735.2	4725.0	4861.3
	2	4718.8	4721.0	4855.2
	3	4692.1	4717.1	4728.3
	**4**	**4561.6**	**4683.8**	**4712.9**
DWG	1	−6426.9	−6549.3	−6575.8
	2	−6543.3	−6602.7	−6591.4
	3	−6566.5	−6603.5	−6600.2
	**4**	**−6597.0**	**−6610.6**	**−6615.1**
YW	1	7610.2	7610.1	7743.2
	2	7608.8	7608.8	7734.2
	3	7605.3	7603.8	7602.9
	**4**	**7583.0**	**7517.1**	**7575.9**

Note: Bold indicates the best model.

**Table 4 animals-15-01733-t004:** The chi-square test of the likelihood ratio values among the different animal models.

Model	df	BW			WW			DGW			YW		
		ABLUP	GBLUP	ssGBLUP	ABLUP	GBLUP	ssGBLUP	ABLUP	GBLUP	ssGBLUP	ABLUP	GBLUP	ssGBLUP
1:2	1	26.46 **	25.28 **	4.70 *	16.38 **	4.00 *	6.09 *	116.37 **	53.41 **	15.68 **	1.37 ^ns^	1.26 ^ns^	8.96 **
1:3	2	30.65 **	25.30 **	32.54 **	43.07 **	7.85 *	133.03 **	139.56 **	54.18 **	24.46 **	4.82 ^ns^	6.30 *	140.27 **
1:4	3	38.11 **	31.31 **	39.00 **	173.57 **	41.20 **	148.41 **	170.08 **	61.30 **	39.30 **	27.18 **	92.98 **	167.32 **
2:3	1	4.19 *	0.01 ^ns^	27.84 **	26.69 **	3.85 *	126.94 **	23.19 **	0.77 ^ns^	8.78 **	3.45 ^ns^	5.04 **	131.31 **
2:4	2	11.65 **	6.02 *	34.29 **	157.19 **	37.20 **	142.32 **	53.70 **	7.89 *	23.62 **	25.81 **	91.73 **	158.36 **
3:4	1	7.46 **	6.01 *	6.45 *	130.52 **	33.35 **	15.38 **	30.52 **	7.12 **	14.84 **	22.35 **	86.68 **	27.05 **

Note: ** means the difference is highly significant (*p* < 0.01); * means the difference is significant (*p* < 0.05); ^ns^ means the difference is not significant (*p* > 0.05). BW: birth weight, WW: weaning weight, DWG: daily weight gain before weaning, YW: yearling weight.

**Table 5 animals-15-01733-t005:** Estimation of the variance components and genetic parameters of birth weight using different methods under the optimal model.

Methods	$\sigma_{a}^{2}$	$\sigma_{m}^{2}$	$\sigma_{a,m}$	$\sigma_{c}^{2}$	$\sigma_{e}^{2}$	$\sigma_{p}^{2}$	$h_{a}^{2}$ ± *SE*	$h_{m}^{2}$	*C* ^2^	$h_{T}^{2}$	*r* _*a*,*m*_
ABLUP	0.02	0.03	−0.01	7.91 × 10^−6^	0.12	0.16	0.10 ± 0.07	0.19	4.94 × 10^−5^	0.13	−0.41
GBLUP	0.01	0.01	7.93 × 10^−4^	3.67 × 10^−9^	0.13	0.15	0.09 ± 0.05	0.07	2.45 × 10^−8^	0.11	0.08
ssGBLUP	0.02	0.01	−4.93 × 10^−3^	0.02	0.12	0.17	0.11 ± 0.06	0.06	0.12	0.10	−0.35

Note:$\;\sigma_{a}^{2}$: direct additive genetic variance; $\sigma_{m}^{2}$: maternal additive genetic variance; $\sigma_{a,m}$: covariance of the individual and maternal genetic effects; $\sigma_{c}^{2}$: maternal permanent environment variance; $\sigma_{e}^{2}$: residual variance;$\;\sigma_{p}^{2}$: phenotypic variance; $h_{a}^{2}$: direct heritability; $h_{m}^{2}$: maternal heritability; *C*^2^: the ratio of the maternal environmental effects variance to the phenotypic variance; $h_{T}^{2}$: total heritability; *r*_*a*,*m*_ is the correlation coefficient between the additive and maternal genetic effects.

**Table 6 animals-15-01733-t006:** Estimation of the variance components and genetic parameters of the weaning weight using different methods under the optimal model.

Methods	$\sigma_{a}^{2}$	$\sigma_{m}^{2}$	$\sigma_{a,m}$	$\sigma_{c}^{2}$	$\sigma_{e}^{2}$	$\sigma_{p}^{2}$	$h_{a}^{2}$ ± *SE*	$h_{m}^{2}$	*C* ^2^	$h_{T}^{2}$	*r* _*a*,*m*_
ABLUP	2.25	1.13	−1.44	3.24 × 10^−5^	3.25	5.19	0.43 ± 0.13	0.22	6.24 × 10^−6^	0.13	−0.91
GBLUP	0.90	0.70	−0.43	9.79 × 10^−8^	3.98	5.15	0.17 ± 0.10	0.14	1.90 × 10^−8^	0.12	−0.54
ssGBLUP	1.23	0.77	−0.74	6.82 × 10^−5^	3.75	5.01	0.25 ± 0.05	0.15	1.36 × 10^−5^	0.10	−0.76

Note:$\;\sigma_{a}^{2}$: direct additive genetic variance; $\sigma_{m}^{2}$: maternal additive genetic variance; $\sigma_{a,m}$: covariance of the individual and maternal genetic effects; $\sigma_{c}^{2}$: maternal permanent environment variance; $\sigma_{e}^{2}$: residual variance; $\sigma_{p}^{2}$: phenotypic variance; $h_{a}^{2}$ direct heritability; $h_{m}^{2}$: maternal heritability; *C*^2^: the ratio of the maternal environmental effects variance to the phenotypic variance; $h_{T}^{2}$: total heritability; *r*_*a*,*m*_ is the correlation coefficient between the additive and maternal genetic effects.

**Table 7 animals-15-01733-t007:** Estimation of the variance components and genetic parameters of the daily weight gain using different methods under the optimal model.

Methods	$\sigma_{a}^{2}$	$\sigma_{m}^{2}$	$\sigma_{a,m}$	$\sigma_{c}^{2}$	$\sigma_{e}^{2}$	$\sigma_{p}^{2}$	$h_{a}^{2}$ ± *SE*	$h_{m}^{2}$	*C* ^2^	$h_{T}^{2}$	*r* _*a*,*m*_
ABLUP	3.38 ×10^−5^	2.97 ×10^−5^	−1.15 × 10^−5^	2.08 × 10^−6^	2.24 × 10^−4^	2.78 × 10^−4^	0.12 ± 0.05	0.11	7.49 × 10^−3^	0.11	−0.37
GBLUP	4.05 ×10^−5^	2.68 ×10^−5^	−1.44 × 10^−5^	2.37 × 10^−6^	2.64 × 10^−4^	3.23 × 10^−4^	0.14 ± 0.04	0.08	7.34 × 10^−3^	0.11	−0.41
ssGBLUP	5.78 ×10^−5^	3.06 ×10^−5^	−1.34 × 10^−5^	2.21 × 10^−6^	3.02 × 10^−4^	3.79 × 10^−4^	0.15 ± 0.05	0.08	5.84 × 10^−3^	0.14	−0.32

Note:$\;\sigma_{a}^{2}$: direct additive genetic variance; $\sigma_{m}^{2}$: maternal additive genetic variance; $\sigma_{a,m}$: covariance of the individual and maternal genetic effects;$\sigma_{c}^{2}$: maternal permanent environment variance; $\sigma_{e}^{2}$: residual variance; $\sigma_{p}^{2}$: phenotypic variance; $h_{a}^{2}$: direct heritability; $h_{m}^{2}$: maternal heritability; *C*^2^: the ratio of the maternal environmental effects variance to the phenotypic variance; $h_{T}^{2}$: total heritability; *r*_*a*,*m*_ is the correlation coefficient between the additive and maternal genetic effects.

**Table 8 animals-15-01733-t008:** Estimation of the variance components and genetic parameters of the yearling weight using different methods under the optimal model.

Methods	$\sigma_{a}^{2}$	$\sigma_{m}^{2}$	$\sigma_{a,m}$	$\sigma_{c}^{2}$	$\sigma_{e}^{2}$	$\sigma_{p}^{2}$	$h_{a}^{2}$ ± *SE*	$h_{m}^{2}$	*C* ^2^	$h_{T}^{2}$	*r* _*a*,*m*_
ABLUP	6.01	4.94	−4.58	3.68 × 10^−4^	12.30	18.67	0.32 ± 0.09	0.26	1.97 × 10^−5^	0.09	−0.84
GBLUP	4.33	2.57	−2.13	3.67 × 10^−4^	16.50	21.27	0.20 ± 0.07	0.12	1.73 × 10^−5^	0.11	−0.64
ssGBLUP	5.31	2.98	−2.24	3.64 × 10^−4^	17.10	23.16	0.23 ± 0.03	0.13	1.57 × 10^−5^	0.15	−0.56

Note:$\;\sigma_{a}^{2}$: direct additive genetic variance; $\sigma_{m}^{2}$: maternal additive genetic variance; $\sigma_{a,m}$: covariance of the individual and maternal genetic effects; $\sigma_{c}^{2}$: maternal permanent environment variance; $\sigma_{e}^{2}$: residual variance; $\sigma_{p}^{2}$: phenotypic variance; $h_{a}^{2}$: direct heritability; $h_{m}^{2}$: maternal heritability; *C*^2^: the ratio of the maternal environmental effects variance to the phenotypic variance; $h_{T}^{2}$: total heritability; *r*_*a*,*m*_ is the correlation coefficient between the additive and maternal genetic effects.

**Table 9 animals-15-01733-t009:** Evaluation of the accuracy and reliability of the estimated genomic breeding values for early growth traits in the three methods.

Traits	Methods	Accuracy	Accuracy Error	Reliability
BW	ABLUP	0.53 ^c^	0.128	0.40
	GBLUP	0.56 ^b^	0.008	0.41
	ssGBLUP	0.61 ^a^	0.006	0.43
WW	ABLUP	0.68 ^b^	0.165	0.50
	GBLUP	0.68 ^b^	0.011	0.51
	ssGBLUP	0.70 ^a^	0.004	0.52
DWG	ABLUP	0.59 ^c^	0.160	0.48
	GBLUP	0.66	0.020	0.53
	ssGBLUP	0.70 ^a^	0.008	0.58
YW	ABLUP	0.60 ^c^	0.149	0.41
	GBLUP	0.64 ^b^	0.021	0.52
	ssGBLUP	0.67 ^a^	0.007	0.59

Note: the same letters indicate no significant difference, while different letters denote significant differences.

## Data Availability

The data that support the findings of this study are available from the corresponding authors upon reasonable request.

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
