# Peer review of "Genomic Selection for Early Growth Traits in Inner Mongolian Cashmere Goats Using ABLUP, GBLUP, and ssGBLUP Methods"

_animals, 2025, doi:10.3390/ani15121733_

Round 1
Reviewer 1 Report
Comments and Suggestions for Authors
Main
The article needs further literature search related to the studied traits in goat species:
- Introduction: You use references from other species: cattle (4), sheep (3), yak (2), pig (2), poultry (1) and human (1). You have only ONE reference in goats, but related to hair follicle growth in Cashmere. I do not understand why you do not cite other more related papers.
- Discussion: Something similar happens in the discussion: sheep (13), cattle (4), alpaca (1), pig (1) and turkey (1). You have included 6 goats references (one of them is related to fleece traits).
The discussion is very superficial, it does not go into depth in trying to explain the results by themselves or in comparison with other similar works. It makes some sense to use papers from other species only when you are discussing methodologies.
Suggestions
General:
- 0.25 is just as informative as 0.253402167. You only need two informative digits in most of the cases: 0.25, 0.00031, 0.016, 14, (1200, 143, without decimal digits)…
- In goats, it is better to say kid than lamb (lines 58 and 60)
Line 15. Change ‘genetic improvement’ for ‘genomic breeding values’ or ‘genetic parameters’ or both
Lines 34 and 122. You have used a GLM procedure previously to the mixed models. The predictions of the fixed effects are not the same. Why do not you predict fixed effects using the best animal model?
Line 42. Your best model might not to be ‘the optimal’. There are other possible models. For example: Model 5. Y = Xb + Z1 a + Z2 m + e (cov(a,m) ≠0)
Change ‘optimal’ for ‘best’
Line 43. If ssGBLUP was the best method, why do you show the heritabilites of all methods? It would be simpler to write 0.11, 0.25, 0.15 and 0.23
Line 44. You only need two informative digits. (0.61-0.70)
Lines 69, 72, 82, … You change the format for referencing citations. Do you prefer Smith et al [##] or Smith [##) et al. ?
Lines 134 and 135. I think it is more understandable to express it as: L134 cov(a , m) = 0 and L135 cov(a , m) ≠ 0
Lines 142-157. Either you omit the definition of the model comparison criteria or you must define them fully and correctly.
Line 147. number of ‘variance’ components
Line 149. You do not define ‘n’. I think it is better – 2logL than – 2l(L).
Line 151. The letter ‘k’ is missing
Line 154. The LR distribution and how the degrees of freedom are obtained is not indicated.
Lines 159-175. You omit the definition of several terms
Line 166. Review Z’Z. You do not define Pi
Line 168. You do not define 𝜎2𝜇
Line 173. You have the matrix [A11 A12 || A21 A22] undefined
Line 175. You have not defined parameters 𝜏, 𝑤, 𝑎, 𝑏, and 𝜔
Line 178. You can delete ‘1’. Suggested wording (lines 177 and 178): The research population was divided into five groups, four of which were combined as the reference group and the remaining as the validation group.
Line 185 cov(a,p)
Linea 187 How do you obtain reliability values?
Table 1
Decide: Number or No.
Mean/SD/C.V: one of the three columns does not provide new information. Remove one of the columns.
DGW: unit is kg/d
Table 2. It is not very informative. You can substitute the table, because you can simply argue that ‘all non-genetic factors were significant’. It might be more informative to know the least square means at each level of each effect.
Line 217. Add ‘test’ to ‘likelihood ratio’
Line 221. You perform chi-square tests to evaluate likelihood ratio test. You have not entered (L154) the distribution and the degrees of freedom to do it.
Table 3. Do you think it is important to keep a decimal place in the values? I think not.
Table 4. Please review these values of chi-square:
WW GBLUP Model 2:3 = 3.85
YW ABLUP Model 1:3 = 4.82
You only need two informative digits
Line 236. Change ‘Table 5-8’ for ’tables’
Line 236. Add ‘(Table 5)’ to ‘BW’
Line 237. You confound additive heritability and total heritability. The values are 0.09-0.11.
Line 241. Add ‘(Table 6)’ to ‘WW’
Line 244, 248 and 253. Delete ‘interaction of’. I suggest ‘covariance between’ or ‘the relationship between’
Line 245. ‘were from -0.54 to -0.91’
Line 245. Add ‘(Table 7)’ to ‘DGW’
Line 246. The values are 0.12-0.15. Delete ‘0.11-0.14’
Line 250. ‘were from -0.41 to -0.32’
Line 250. Add ‘(Table 8)’ to ‘YW’
Line 251. You confound additive heritability and total heritability. The values are 0.23-0.32.
Line 253. ‘was from 1.6 x 10-5 to 2.0 x 10-5’
Lines 254-255. ‘were from -0.84 to-0.56’
Tables 5-8. No reference is made to the estimated variance components. You can remove these six columns from the tables. There is no indication of how the total heritability has been calculated. I assume (var(a)+var(m)+ 2cov(a,m)) / var(P))
Lines 299-300. For BW, accuracy of ABLUP and GBLUP methods were not different.
Table 9. You only need two informative digits
Line 310. I would delete ‘extremely or highly’
Lines 314, 315 and 319. I would delete ‘highly’
Line 316. Author: Jalil-Sarghale
Lines 323-330. Are you referring to your results or those of Lupi et al.(2016)? Kids and goats or lambs and ewes?
Line 327. ‘Within’ in lower case after comma.
Line 328. Change ‘better’ for ‘higher’
Line 334. Change ‘interaction’ for ‘covariance between’
Line 338. Citation error: Kumar el al. [34].
Lines 338-340. Unconnected sentence ‘Kumar et al. ….in other seasons’
Line 344. Add ‘et al.’
Lines 346-347.You can delete ‘and fleece’
Line 347. Citation error: Dege [39]
Line 347. Inconsistent citation: Dige et al. is related to growth and feed efficiency in goats.
Line 351. Citation error: Ulutas [40]. Which models did he compare?
Line 355. You confound heritability and total heritability values
Line 357. Citation error: Di [42].
Lines 359, 360 and 363. You confound heritability and total heritability values
Line 363. Add ‘et al.’
Line 365. [46] Angora goats and [47] Black Bengal goats. Please, rephrase ‘Angora goats Black goats’.
Lines 376 and 377. Change ‘methods’ for ‘method’
Line 378. Citation error: Shiavas et al. [50]
Lines 391-393. Suggestion: ‘For genetic evaluation of early growth traits in IMCG, animal models should include as rando the direct additive genetic effects, the maternal genetic effects, the maternal environmental effects and the covariance between direct additive and maternal genetic effects.’
Lines 397 399. Genetic progress does not depend on heritability alone. Selection intensity and observed variation must be considered, as well as the possible reduction of the generation interval using genomic selection.
Line 398. Change 0.23 for 0.20
Line 400. It is redundant to say that it improves the precision of genomic values(GEBV) when genomic selection is performed. It is the model used that improves accuracy. In this case, the best model was ssGBLUP. I assume you meant EBV and not ‘GEBV’.
References
1. The refrence is repeated
5, 6, 11, 14, 15, 16, 20, 22, 28. Journal name
32. Replace ‘Segureo’ for ‘Segureño’
34. Uncited. Journal name.
38, 41, 44, 45, 48, 49. Journal name
40. Uncited. Journal name
42. Uncited
50. Uncited
Comments on the Quality of English Language- One space before opening with ‘[‘
- One space after closing with ‘]’
- (In the text) One space after points or commas
- One space after colons
- Line 66. Suggestion: change ‘provided’ to ‘proposed’
- Lines 76 and 77. Please, rephrase the sentence because it is not understood
- Line 183. Delete ‘accuracy result’
- Line 245. Add ‘values’ after the adjective negative
- Line 245. Delete ‘respectively’
- Line 250. Delete ‘respectively’
- Line 255 Delete ‘respectively’
- Line 307. A connector between ‘dams’ and ‘herds’ is missing: ‘and’ or ‘or’
Author Response
I deeply admire your solid expertise in breeding and statistical methodology. Your systematic review of the manuscript and invaluable suggestions have significantly improved its quality. I sincerely apologize for the delayed response, as I was occupied with submitting my graduation thesis over the past month. Nevertheless, I’m truly grateful for your constructive feedback. I kindly request your further review of the revised manuscript. Thank you very much for your time and guidance. I have carefully addressed all your comments and revised the manuscript accordingly. I hope this version shows significant improvement.
The article needs further literature search related to the studied traits in goat species:
- Introduction: You use references from other species: cattle (4), sheep (3), yak (2), pig (2), poultry (1) and human (1). You have only ONE reference in goats, but related to hair follicle growth in Cashmere. I do not understand why you do not cite other more related papers.
- Discussion: Something similar happens in the discussion: sheep (13), cattle (4), alpaca (1), pig (1) and turkey (1). You have included 6 goats references (one of them is related to fleece traits).
The discussion is very superficial, it does not go into depth in trying to explain the results by themselves or in comparison with other similar works. It makes some sense to use papers from other species only when you are discussing methodologies.
Response: Thanks I agree with your viewpoint. However, there are relatively few literature reports on genome selection of goats. There are some reports on the milk production traits of dairy goats and the fleece traits of cashmere goats, while there are almost no reports on genome selection of early growth traits. So I have to use the reference from other species.
Suggestions
General:
- 0.25 is just as informative as 0.253402167. You only need two informative digits in most of the cases: 0.25, 0.00031, 0.016, 14, (1200, 143, without decimal digits)…
Response: I agree with your opinion. We have revised them in the revised version.
- In goats, it is better to say kid than lamb (lines 58 and 60)
Response: Thanks for your advice. I have updated it in the revised manuscript.
Line 15. Change ‘genetic improvement’ for ‘genomic breeding values’ or ‘genetic parameters’ or both
Response: Thanks for your advice. I have updated it in the revised manuscript.
Lines 34 and 122. You have used a GLM procedure previously to the mixed models. The predictions of the fixed effects are not the same. Why do not you predict fixed effects using the best animal model?
Response: Generally, the phenotypic value of a trait is influenced by both fixed effects (systematic environmental effects) and random effects (additive genetic effects and permanent environmental effects). Systematic environmental effects include factors such as the year of measurement, herds, maternal age, etc. Therefore, the model in line 122 is used to determine which systematic environmental effects influence the phenotypic value. However, models 1-6 are used to determine which random effects influence the phenotypic value. Typically, the sequence of genetic evaluation work involves first identifying the systematic environmental effects and, based on this, then determining the random effects, i.e., selecting the best model.
Line 42. Your best model might not to be ‘the optimal’. There are other possible models. For example: Model 5. Y = Xb + Z1 a + Z2 m + e (cov(a,m) ≠0)
Change ‘optimal’ for ‘best’
Response: I agree with your opinion. It is a very good view. Our team recently conducted a genetic evaluation of early growth traits using another dataset with six models. Indeed, the model you suggested proved to be the optimal one. However, excluding the maternal permanent environmental effect did not have a significant impact on other results, including the estimation of genetic parameters or the accuracy of breeding values.
Line 43. If ssGBLUP was the best method, why do you show the heritabilites of all methods? It would be simpler to write 0.11, 0.25, 0.15 and 0.23
Response: I have rewritten this part. Hope that it looks better.
Line 44. You only need two informative digits. (0.61-0.70)
Response: Yes, we have updated them. Thanks.
Lines 69, 72, 82, … You change the format for referencing citations. Do you prefer Smith et al [##] or Smith [##) et al. ?
Response: I prefer to Smith et al [##] I have updated in the whole text.
Lines 134 and 135. I think it is more understandable to express it as: L134 cov(a , m) = 0 and L135 cov(a , m) ≠ 0
Response: Thanks for your reminder. I have updated them in the revised version.
Lines 142-157. Either you omit the definition of the model comparison criteria or you must define them fully and correctly.
Response: I indicated the definition of the model comparison criteria with LRT in the revised version.
Line 147. number of ‘variance’ components
Response: Thanks. I have revised it.
Line 149. You do not define ‘n’. I think it is better – 2logL than – 2l(L).
Response: Thanks for your reminder. This is a mistake.
Line 151. The letter ‘k’ is missing
Response: Thanks for your reminder. I have added it in the revised version.
Line 154. The LR distribution and how the degrees of freedom are obtained is not indicated.
Response: Thanks for your suggestions. I have added it in the revised version.
Lines 159-175. You omit the definition of several terms.
Response: Thanks for your suggestions. Another reviewer suggest that one method for comparing the models was remained. So I have deleted the description for AIC and BIC methods.
Line 166. Review Z’Z. You do not define Pi
Response: I have added this definition for Pi
Line 168. You do not define ?2?
Response: Sorry, this is a mistake, I have replaced ?2? with ?2a.
Line 173. You have the matrix [A11 A12 || A21 A22] undefined
Response: I have added the definition of A11, A12,A21 and A22.
Line 175. You have not defined parameters ?, ?, ?, ?, and ?
Response: I have updated the formula for H-1.
Line 178. You can delete ‘1’. Suggested wording (lines 177 and 178): The research population was divided into five groups, four of which were combined as the reference group and the remaining as the validation group.
Response: Thanks so much for your advice. I have updated this sentence.
Line 185 cov(a,p)
Response: Thanks for you correction. You are truly a conscientious reviewer. Thanks so much!
Linea 187 How do you obtain reliability values?
Response: I have added the formula of reliability values. Hope that it is OK.
Table 1
Decide: Number or No.
Mean/SD/C.V: one of the three columns does not provide new information. Remove one of the columns.
DGW: unit is kg/d
Response: I agree with your view. I have incorporated the updates mentioned above.
Table 2. It is not very informative. You can substitute the table, because you can simply argue that ‘all non-genetic factors were significant’. It might be more informative to know the least square means at each level of each effect.
Response: I agree with your suggestion. If I were to list all the contents of the ANOVA table for each trait, including degrees of freedom, sum of squares, mean square, F-value and P-value, the table would be very complex. So I would list the core values of F and P. I hope that it is OK.
Line 217. Add ‘test’ to ‘likelihood ratio’
Response: Yes, I have added it.
Line 221. You perform chi-square tests to evaluate likelihood ratio test. You have not entered (L154) the distribution and the degrees of freedom to do it.
Response: I have made an detailed explanation in the revised version.
Table 3. Do you think it is important to keep a decimal place in the values? I think not.
Table 4. Please review these values of chi-square:
WW GBLUP Model 2:3 = 3.85
YW ABLUP Model 1:3 = 4.82
You only need two informative digits
Response: Your statistical expertise is truly outstanding. I have carefully verified each value and reassigned the asterisks (*) to indicate significance levels. Hope that the revised version is OK. Thank you for your valuable reminders and guidance.
Line 236. Change ‘Table 5-8’ for ’tables’
Response: I have updated this.
Line 236. Add ‘(Table 5)’ to ‘BW’
Response: I have added Table 5 to BW.
Line 237. You confound additive heritability and total heritability. The values are 0.09-0.11.
Response: Thanks. I have updated in the whole text.
Line 241. Add ‘(Table 6)’ to ‘WW’
Response: I have added Table 6 to WW.
Line 244, 248 and 253. Delete ‘interaction of’. I suggest ‘covariance between’ or ‘the relationship between’
Response: I have replaced “interaction of” with “covariance between ”.
Line 245. ‘were from -0.54 to -0.91’
Response: thanks so much for carefully check for all of the digits.
Line 245. Add ‘(Table 7)’ to ‘DGW’
Response: I have added Table 8 to DGW.
Line 246. The values are 0.12-0.15. Delete ‘0.11-0.14’
Response: I have updated this. Thanks.
Line 250. ‘were from -0.41 to -0.32’
Response: I have updated this. Thanks.
Line 250. Add ‘(Table 8)’ to ‘YW’
Response: I have added Table 8 to YW.
Line 251. You confound additive heritability and total heritability. The values are 0.23-0.32.
Line 253. ‘was from 1.6 x 10-5 to 2.0 x 10-5’
Lines 254-255. ‘were from -0.84 to-0.56’
Response: Many thanks for your thorough review. All the aforementioned points have been updated accordingly.
Tables 5-8. No reference is made to the estimated variance components. You can remove these six columns from the tables. There is no indication of how the total heritability has been calculated. I assume (var(a)+var(m)+ 2cov(a,m)) / var(P))
Response: I have added the formula of total heritability in the revised version. And explain the variance components for the random effect.
Lines 299-300. For BW, accuracy of ABLUP and GBLUP methods were not different.
Response: The significant differences detected in the multiple comparisons may be related to the larger error values in ABLUP.
Table 9. You only need two informative digits
Response: I have remained two informative digits for accuracy and reliability. Due to some accuracy errors being too small, three digits have been retained.
Line 310. I would delete ‘extremely or highly’
Response: I agree with your opinion. I have deleted ‘extremely or highly’.
Lines 314, 315 and 319. I would delete ‘highly’
Response: I agree with your opinion. I have deleted ‘highly’.
Line 316. Author: Jalil-Sarghale
Response: Thanks for your reminder. I have revised this.
Lines 323-330. Are you referring to your results or those of Lupi et al.(2016)? Kids and goats or lambs and ewes?
Response: Thanks for your reminder. This is our results. I have updated this sentence. Hope that it looks better.
Line 327. ‘Within’ in lower case after comma.
Response: Thanks for your reminder. I have updated.
Line 328. Change ‘better’ for ‘higher’
Response: Thanks for your reminder. I have updated.
Line 334. Change ‘interaction’ for ‘covariance between’
Response: Thanks for your opinion. I have updated.
Line 338. Citation error: Kumar el al. [34].
Response: Thanks for your reminder. I have updated this. Thank you.
Lines 338-340. Unconnected sentence ‘Kumar et al. ….in other seasons’
Response: Thanks for your reminder. I have updated this sentence.
Line 344. Add ‘et al.’
Response: Thanks for your reminder. I have updated this.
Lines 346-347.You can delete ‘and fleece’
Response: Thanks for your reminder. I have deleted it.
Line 347. Citation error: Dege [39]
Response: Thanks for your reminder. I have updated this.
Line 347. Inconsistent citation: Dige et al. is related to growth and feed efficiency in goats.
Response: Thanks for your reminder. I have updated this.
Line 351. Citation error: Ulutas [40]. Which models did he compare?
Response: I have updated this sentence. Hope that it looks better.
Line 355. You confound heritability and total heritability values
Response: I have updated it. Hope that it looks better.
Line 357. Citation error: Di [42].
Response: I have updated it.
Lines 359, 360 and 363. You confound heritability and total heritability values
Response: Thanks for your reminder. I have updated these in the whole text.
Line 363. Add ‘et al.’
Response: I have added it. Thanks.
Line 365. [46] Angora goats and [47] Black Bengal goats. Please, rephrase ‘Angora goats Black goats’.
Response: I have updated this. Thanks.
Lines 376 and 377. Change ‘methods’ for ‘method’
Response: I have updated this. Thanks.
Line 378. Citation error: Shiavas et al. [50]
Response: I have updated this.
Lines 391-393. Suggestion: ‘For genetic evaluation of early growth traits in IMCG, animal models should include as rando the direct additive genetic effects, the maternal genetic effects, the maternal environmental effects and the covariance between direct additive and maternal genetic effects.’
Response: Thanks for your suggestions. I have updated this.
Lines 397 399. Genetic progress does not depend on heritability alone. Selection intensity and observed variation must be considered, as well as the possible reduction of the generation interval using genomic selection.
Response: Thanks for your opinion. I have updated it. Hope that it is OK.
Line 398. Change 0.23 for 0.20
Response: Thanks for your reminder. I have updated this.
Line 400. It is redundant to say that it improves the precision of genomic values(GEBV) when genomic selection is performed. It is the model used that improves accuracy. In this case, the best model was ssGBLUP. I assume you meant EBV and not ‘GEBV’.
Response: Thanks for your reminder. I have updated this sentence. Hope that it looks better.
References
- The refrence is repeated
5, 6, 11, 14, 15, 16, 20, 22, 28. Journal name
- Replace ‘Segureo’ for ‘Segureño’
- Uncited. Journal name.
38, 41, 44, 45, 48, 49. Journal name
- Uncited. Journal name
- Uncited
- Uncited
Response: I have finished the revision above mentioned.

Reviewer 2 Report
Comments and Suggestions for Authors
The paper addresses an important topic in goat breeding: the genetic improvement of early growth traits using genomic information. The study utilizes a relatively low dataset of over 2,256 individuals in a local breed (Inner Mongolia Cashmere goats (IMCGs) with both phenotypic and genotypic data. This analysis explores various factors affecting early growth traits, including both fixed and random effects, and compares different models and methods for genomic selection. The results are presented in a clear and concise manner, with tables and figures that effectively summarize the key findings. The study provides practical recommendations for breeding programs aimed at improving early growth traits in IMCGs.
However, the study is based on a very specific local population of IMCGs, and the findings may not be directly applicable to other goat populations. moreover, the study largely confirms previous findings on the factors affecting early growth traits and the performance of different genomic selection methods.
Nevertheless, the fact that today ssblup is the reference genomic assessment methodology for most species is based on the fact that it allows the inclusion of phenotyped and non-genotyped animals, which are generally much more numerous than those genotyped. In this work, ssblup has not been used in this approach, but a model has been developed in which all animals are genotyped. This fact must be justified, since in practice it is difficult to expect that in a goat breed all phenotyped animals are genotyped. Since the result of ABlup will depend largely on the reliability and depth of the pedigree, it is essential to add information on the quality of said pedigree (number of equivalent and complete generations), mechanisms implemented to ensure the reliability of the paternities, etc. etc. It would also be important to know the correlation between matrices A and G before and after the formation of matrix H.
Finally, a discussion on the practical implications of this study in relation to the genetic improvement of the population studied is missing. in the same way, the authors could discuss these limitations of the study in terms of generalizability and suggest future research directions that could address these limitations.
Some recommendations that could improve the study:
- Since there are some published papers that analyse the environmental and genetic factors that determine the traits analysed. It does not make much sense to include only studies on pig, cattle, sheep or Yak (e.g. Terakado et al , Balasundaram et al or Fei Ge).
- - Quality control: The animals and snps that were available after quality control are not indicated. This could indicate why they have been imputed once the snps that do not meet some criteria such as MAF, HW equilibrium or linkage disequilibrium have been eliminated. In any case, the criteria for eliminating snps and animals must be rewritten to make them understandable. For example, criteria 3 and 4 are not the same and do they overlap with criteria 1 and 2?
- In the case of SNPS in general, the number of markers used is not a critical aspect, but rather the number of animals and how they have been selected (whether or not they are representative of the population under evaluation, whether they have been selected as a reference population based on the relationship with the phenotyped animals, etc. etc.). All these aspects must be clarified and justified.
- - Three methods are used to determine the best model, which seems redundant; the one considered most appropriate to compare the objective of the study (predictive power) and the type of models (nested models with cross-validation) could be used.
- - To facilitate the readability of the papers, it is recommended to eliminate decimals in the text, e.g. 0.5209-0.6057, 0.6751- 248 0.7003,0.5857- 0.7021 and 0.5965-0.6683, the situation is perfectly described by including only 3 decimals.
- In Table 1, a much lower number of DGW data can be observed. If birth and weaning data are available, why are 50% of the DGW data not available?
- The selection of the best model based on theoretical model fit data does not presuppose that it is the optimal model for genetic evaluations. But the estimation that is achieved in the reliability of said genetic values when genomics is used (ssblup and gblup) does not either. In the paper, the higher reliability of ssBLUP is due to the higher h2 obtained compared to genomic selection. Since a cross-validation has been carried out, the predictive value of each model should be used to ensure the improvement in the estimation of the genetic values. It would also be interesting to make some considerations in the variation of reliability according to the reliability of the model without genomics, according to the depth and completeness of the pedigree of the animal, etc.
nothing to comment
Author Response
I deeply admire your solid expertise in breeding. Your systematic review of the manuscript and invaluable suggestions have significantly improved its quality. Additionally, I sincerely apologize for the delayed response, as I was occupied with submitting my graduation thesis over the past month. Nevertheless, I’m truly grateful for your constructive feedback. I kindly request your further review of the revised manuscript. Thank you very much for your time and guidance. I have carefully addressed all your comments and revised the manuscript accordingly. I hope this version shows significant improvement.
The paper addresses an important topic in goat breeding: the genetic improvement of early growth traits using genomic information. The study utilizes a relatively low dataset of over 2,256 individuals in a local breed (Inner Mongolia Cashmere goats (IMCGs) with both phenotypic and genotypic data. This analysis explores various factors affecting early growth traits, including both fixed and random effects, and compares different models and methods for genomic selection. The results are presented in a clear and concise manner, with tables and figures that effectively summarize the key findings. The study provides practical recommendations for breeding programs aimed at improving early growth traits in IMCGs.
However, the study is based on a very specific local population of IMCGs, and the findings may not be directly applicable to other goat populations. moreover, the study largely confirms previous findings on the factors affecting early growth traits and the performance of different genomic selection methods.
Nevertheless, the fact that today ssblup is the reference genomic assessment methodology for most species is based on the fact that it allows the inclusion of phenotyped and non-genotyped animals, which are generally much more numerous than those genotyped. In this work, ssblup has not been used in this approach, but a model has been developed in which all animals are genotyped. This fact must be justified, since in practice it is difficult to expect that in a goat breed all phenotyped animals are genotyped. Since the result of ABlup will depend largely on the reliability and depth of the pedigree, it is essential to add information on the quality of said pedigree (number of equivalent and complete generations), mechanisms implemented to ensure the reliability of the paternities, etc. etc. It would also be important to know the correlation between matrices A and G before and after the formation of matrix H.
Finally, a discussion on the practical implications of this study in relation to the genetic improvement of the population studied is missing. in the same way, the authors could discuss these limitations of the study in terms of generalizability and suggest future research directions that could address these limitations.
Response: Thank you very much for your careful review of our manuscript. We agree with some of the viewpoints you mentioned. Indeed, numerous studies have shown that ssGBLUP is a universal method for important economic traits in livestock. But one of our analysis with another batch of data showed that the accuracy of genomic prediction with GBLUP method for body weight has higher accuracy than that with ssGBLUP method.
Some recommendations that could improve the study:
- Since there are some published papers that analyse the environmental and genetic factors that determine the traits analysed. It does not make much sense to include only studies on pig, cattle, sheep or Yak (e.g. Terakado et al , Balasundaram et al or Fei Ge).
Response: I agree with your viewpoint. There are some reports on the milk production traits of dairy goats and the fleece traits of cashmere goats, while there are almost no reports on genome selection of early growth traits. So I have to use the reference from other species.
- Quality control: The animals and snps that were available after quality control are not indicated. This could indicate why they have been imputed once the snps that do not meet some criteria such as MAF, HW equilibrium or linkage disequilibrium have been eliminated. In any case, the criteria for eliminating snps and animals must be rewritten to make them understandable. For example, criteria 3 and 4 are not the same and do they overlap with criteria 1 and 2?
Response: I agree with your opinion. I have rewritten this section. Hope that it looks better.
- In the case of SNPS in general, the number of markers used is not a critical aspect, but rather the number of animals and how they have been selected (whether or not they are representative of the population under evaluation, whether they have been selected as a reference population based on the relationship with the phenotyped animals, etc. etc.). All these aspects must be clarified and justified.
Response: Thanks for your suggestion. Since animal population in our study is not particularly large, five-fold cross validation may be an ideal method for assessing the accuracy of GEBV prediction.
- - Three methods are used to determine the best model, which seems redundant; the one considered most appropriate to compare the objective of the study (predictive power) and the type of models (nested models with cross-validation) could be used.
Response: I agree with your opinion. I have deleted AIC and BIC methods in the revised version.
- - To facilitate the readability of the papers, it is recommended to eliminate decimals in the text, e.g. 0.5209-0.6057, 0.6751- 248 0.7003,0.5857- 0.7021 and 0.5965-0.6683, the situation is perfectly described by including only 3 decimals.
Response: Thanks for your suggestion. Another reviewer also raised the same opinion. I have remained two digits for the accuracy and realibity.
- In Table 1, a much lower number of DGW data can be observed. If birth and weaning data are available, why are 50% of the DGW data not available?
Response: the specific birth dates for some individuals are not recorded, so there are fewer records of daily weight gain.
- The selection of the best model based on theoretical model fit data does not presuppose that it is the optimal model for genetic evaluations. But the estimation that is achieved in the reliability of said genetic values when genomics is used (ssblup and gblup) does not either. In the paper, the higher reliability of ssBLUP is due to the higher h2 obtained compared to genomic selection. Since a cross-validation has been carried out, the predictive value of each model should be used to ensure the improvement in the estimation of the genetic values. It would also be interesting to make some considerations in the variation of reliability according to the reliability of the model without genomics, according to the depth and completeness of the pedigree of the animal, etc.
Response: I agree with your viewpoint. Thanks for your careful review. We will continuously expand the reference population of Inner Mongolia cashmere goats, carry out genome selection for individuals, and apply it to genetic breeding for superior individuals.

Reviewer 3 Report
Comments and Suggestions for Authors
The authors compared different statistical models to estimate breeding values for early growth traits in Cashmere goats. Birth year, herd, sex, birth type, and dam age significantly influenced these traits. The best model included additive genetic effects of the individual, maternal genetic effects, and environmental effects. Among the methods tested, ssGBLUP provided the most accurate breeding value estimates. The manuscript contains interesting results but requires considerable improvement in grammar and clarity. The authors must work with English language experts to rewrite their paper. The manuscript is difficult to evaluate because the tables are missing.
Lines 19-20: “researchers” and “they” indicate you are referring to someone else’s research, not your work.
Line 32: Delete the spaces before the colon.
Line 33: Change to “daily weight gain”.
Lines 37-38: Change to “was used to assess”.
Line 39: Delete “It showed that”.
Lines 39-40: “and other traits” is too vague. List the traits to improve the clarity of the statement.
Line 43: Change the acronym to DWG since the correct name of the trait is daily weight gain (see line 33).
Lines 50-52: Change to "The Inner Mongolia Cashmere goat (IMCG), a highly regarded breed known worldwide, has been developed over a long period through natural selection and artificial breeding for both cashmere and meat production."
Line 54: “goat breeds” instead of “goats breeds”.
Line 55: Change to “…as one of the first breeds…”.
Line 57: Change to “Early growth traits have strong implications for…”.
Lines 58-60: Change to "The growth and development of young lambs are directly linked to the economic benefits of livestock production, serving as key indicators of growth rate, health status, and overall productivity."
Line 60: Delete “the”.
Lines 66-67: Do not begin a sentence with “And”.
Line 66: “described” instead of “provided”.
Line 67: Change to “…selection of superior animals and plants [8-10]”.
Lines 68-74: Change to “Lavvaf and Noshary used a single-trait animal model to perform genetic evaluation of early growth traits in the Lori breed of sheep. Early growth traits were significantly affected by direct additive genetic effects, direct maternal genetic effects, and environmental effects [11]. Balasundaram et al. evaluated the genetic parameters of growth traits and quantitative genetic metrics of Mecheri sheep in Tamil Nadu. The best model included direct and maternal genetic effects as random effects with no covariance [12]”.
Where is Tamil Nadu? Is it in China?
Lines 78-79: “yak” or “Yak”? Be consistent.
Line 94: “data were” instead of “data was”. The word “data” is plural.
Line 96: “dairy gain weight”?
Line 105: Normally that is the case. Was that the criterion for outliers in your study?
Lines 106-107: “Then the genotype data were performed quality control” does not make sense. Please reword.
Line 108: Leave a space in “than95%”.
Line 110: What is “deletion rate”? Are you referring to InDels?
Line 112: Delete “at the sample individuals”.
Line 113: Leave a space after the semicolon.
Line 115: Delete “were”.
Line 116: Change the subheading to “Determination of fixed and random effects “.
Line 118: “was” instead of “were”.
Line 119: This is the first mention of “herds”. Describe the herds earlier in the Materials and Methods. How were the herds chosen for your study?
Line 120: “triplets” instead of “triple”.
Lines 124-128: Use superscripts and subscripts in the appropriate places.
Line 127: Change to “herd, Dl is the effect of the lth age of dam…”.
Lines 134-135: This would be clearer if you used cov (a,m) = 0 and cov (a,m) ≠ 0.
Lines 140-141: Change to “…is the presence or absence of covariance between…”.
Lines 142-144: Change to “The most appropriate model was obtained using the Akaike Information Criterion (AIC), the Bayesian information criteria (BIC) and the likelihood ratio test (-2LogL) [20]. The formula used to calculate AIC was as follows:”.
Line 148: Change to “The formula used to calculate BIC was…”.
Lines 150-152: The symbols are missing in this statement.
Line 153: Change to “The formula used to calculate the likelihood ratio value was as follows:”.
Lines 146, 150, and 155: “where” or “Where”? Be consistent.
Lines 163 and 165: “using” instead of “by”.
Line 178: Delete “1”.
Line 179: Do not begin a sentence with “And”. Change “5” to “five”.
Line 183: It does not make sense to say the accuracy is more accurate. Reword.
Line 185: Add a comma in “cov (a p)”.
Line 191: “are shown” rather than “was shown”. I do not see Table 1.
Lines 192-193: Change to “…and 37.20 kg, respectively, and the coefficients of variation ranged from…”.
Line 197: I do not see Table 2.
Lines 198-205: P < 0.05 indicates significant differences between means. P < 0.01 shows highly significant differences between means. P < 0.001 does not have a description such as extremely significant.
Line 208: Where is Table 3?
Line 221: I cannot find Tables 5 – 8.
Lines 224, 229, 233, and 238: “interaction” or “correlation”?
Line 247: I cannot find Table 9.
Line 251: Do not begin a sentence with “And”.
Line 256: Delete “so”.
Line 257: Insert “and” before “herds”.
Line 260: Delete “extremely or”.
Lines 271-273: This is an incomplete sentence and must be rewritten.
Line 273: Delete “It was obtained that”.
Lines 276-277: Why would the reproductive ability of dams influence early growth rate? What do you mean by ewes’ body function? Be more specific.
Line 277: “within” does not need to be capitalized.
Lines 283-285: Here is a clearer and more grammatically correct version of your sentence:
"In this study, four animal models were constructed to assess the maternal genetic effect, maternal environmental effect, and the covariance between the direct additive effect and maternal genetic effect."
Line 287: Change “interaction” to “covariance”.
Lines 291-292: This is an incomplete sentence. Reword.
Line 294: “Black” should be capitalized since it is part of the breed’s name. “is” instead of “was”.
Line 295: Change to “…genetic parameters. The results…”.
Line 307: “Merino” needs to be capitalized since it is the name of a breed.
Line 308: “Merinos” must be capitalized since it is part of the breed’s name.
Lines 308-309: “All of which were similar to the results in our study” is an incomplete sentence. Reword.
Line 315: “Angora goats Black goats” does not make sense. What is the correct name of the breed?
Line 316: Leave a space between the sentences. What is “It”?
Line 321: What is “descent determination”?
Line 327: Change to “…was almost as high as…”.
Line 329: Insert “and” before “Bayes”.
Lines 341-343: Change to “Animal models for genetic evaluation of early growth traits in IMCGs should include direct additive genetic effects, maternal genetic effects, maternal environmental effects and the covariance between direct additive and maternal genetic effects”.
Do the maternal permanent environmental effects need to be included in the model given that they accounted for a tiny proportion of the phenotypic variance in your study?
Line 345: Here and throughout the manuscript, change “daily gain weight” to “daily weight gain”.
Lines 360-364: Only use the authors’ initials in this section.
Comments on the Quality of English Language
The manuscript requires considerable improvement in grammar and clarity. The authors must work with English language experts to rewrite their paper.
Author Response
I deeply admire your solid expertise in breeding. Your systematic review of the manuscript and invaluable suggestions have significantly improved its quality. Additionally, I sincerely apologize for the delayed response, as I was occupied with submitting my graduation thesis over the past month. Nevertheless, I’m truly grateful for your constructive feedback. I kindly request your further review of the revised manuscript. Thank you very much for your time and guidance. I have carefully addressed all your comments and revised the manuscript accordingly. I hope this version shows significant improvement.
The authors compared different statistical models to estimate breeding values for early growth traits in Cashmere goats. Birth year, herd, sex, birth type, and dam age significantly influenced these traits. The best model included additive genetic effects of the individual, maternal genetic effects, and environmental effects. Among the methods tested, ssGBLUP provided the most accurate breeding value estimates. The manuscript contains interesting results but requires considerable improvement in grammar and clarity. The authors must work with English language experts to rewrite their paper. The manuscript is difficult to evaluate because the tables are missing.
Response:Thank you very much for your careful review. We have made revisions item by item and returned the document. Hope that it looks better. We sincerely appreciate you taking the time to review it again.
Lines 19-20: “researchers” and “they” indicate you are referring to someone else’s research, not your work.
Response:Thanks for your suggestion. I have updated it in the revised version. Hope that it is OK.
Line 32: Delete the spaces before the colon.
Response:Thanks. I have deleted it.
Line 33: Change to “daily weight gain”.
Response:Thanks. I have changed it.
Lines 37-38: Change to “was used to assess”.
Response:Thanks. I have changed it.
Line 39: Delete “It showed that”.
Response:Thanks. I have deleted it.
Lines 39-40: “and other traits” is too vague. List the traits to improve the clarity of the statement.
Response:I agree with your opinion. I have listed the traits name.
Line 43: Change the acronym to DWG since the correct name of the trait is daily weight gain (see line 33).
Response:Yes, I have updated it in the whole text.
Lines 50-52: Change to "The Inner Mongolia Cashmere goat (IMCG), a highly regarded breed known worldwide, has been developed over a long period through natural selection and artificial breeding for both cashmere and meat production."
Response:Thanks. I have updated it.
Line 54: “goat breeds” instead of “goats breeds”.
Response:I have updated it. Thanks.
Line 55: Change to “…as one of the first breeds…”.
Response:Thanks. I have updated it.
Line 57: Change to “Early growth traits have strong implications for…”.
Response:Thanks. I have updated it.
Lines 58-60: Change to "The growth and development of young lambs are directly linked to the economic benefits of livestock production, serving as key indicators of growth rate, health status, and overall productivity."
Response:Thanks. I have updated it.
Line 60: Delete “the”.
Response:Thanks. I have deleted it.
Lines 66-67: Do not begin a sentence with “And”.
Response:Thanks. I have deleted it.
Line 66: “described” instead of “provided”.
Response:Thanks. I have updated it.
Line 67: Change to “…selection of superior animals and plants [8-10]”.
Response:Thanks. I have updated it.
Lines 68-74: Change to “Lavvaf and Noshary used a single-trait animal model to perform genetic evaluation of early growth traits in the Lori breed of sheep. Early growth traits were significantly affected by direct additive genetic effects, direct maternal genetic effects, and environmental effects [11]. Balasundaram et al. evaluated the genetic parameters of growth traits and quantitative genetic metrics of Mecheri sheep in Tamil Nadu. The best model included direct and maternal genetic effects as random effects with no covariance [12]”.
Response:Thanks. I have updated it.
Where is Tamil Nadu? Is it in China?
Response:No. It is in India.
Lines 78-79: “yak” or “Yak”? Be consistent.
Response:Thanks. I have updated this.
Line 94: “data were” instead of “data was”. The word “data” is plural.
Response:I agree with your opinion. I have updated this.
Line 96: “dairy gain weight”?
Response:Thanks. I have updated it.
Line 105: Normally that is the case. Was that the criterion for outliers in your study?
Response:Yes. We deleted the outliers with this criterion in my study.
Lines 106-107: “Then the genotype data were performed quality control” does not make sense. Please reword.
Response:I have updated this sentence. Hope that it looks better.
Line 108: Leave a space in “than95%”.
Response:Thanks. I have updated it.
Line 110: What is “deletion rate”? Are you referring to InDels?
Response:Thanks. I have updated it. Actually, the deletion rate is the missing genotype rate.
Line 112: Delete “at the sample individuals”.
Response:Thanks. I have deleted it.
Line 113: Leave a space after the semicolon.
Response:Thanks. I have updated it.
Line 115: Delete “were”.
Response:Thanks. I have deleted it.
Line 116: Change the subheading to “Determination of fixed and random effects “.
Response:Thanks. I have updated it.
Line 118: “was” instead of “were”.
Response:Thanks. I have updated it.
Line 119: This is the first mention of “herds”. Describe the herds earlier in the Materials and Methods. How were the herds chosen for your study?
Response:Thanks. The herds were explained in our previously published paper. A total of 12 herds was considered in this study, including 9 female herds and 3 male herds. I have described the herds in the revised version.
Line 120: “triplets” instead of “triple”.
Response:Thanks. I have updated it.
Lines 124-128: Use superscripts and subscripts in the appropriate places.
Response:Thanks. I have updated it.
Line 127: Change to “herd, Dl is the effect of the lth age of dam…”.
Response:Thanks. I have updated it.
Lines 134-135: This would be clearer if you used cov (a,m) = 0 and cov (a,m) ≠ 0.
Response:Thanks. I have updated it.
Lines 140-141: Change to “…is the presence or absence of covariance between…”.
Response:Thanks. I have updated it.
Lines 142-144: Change to “The most appropriate model was obtained using the Akaike Information Criterion (AIC), the Bayesian information criteria (BIC) and the likelihood ratio test (-2LogL) [20]. The formula used to calculate AIC was as follows:”.
Response:Thanks. I have updated it.
Line 148: Change to “The formula used to calculate BIC was…”.
Response:Thanks. I have updated it.
Lines 150-152: The symbols are missing in this statement.
Response:Thanks. I have updated it.
Line 153: Change to “The formula used to calculate the likelihood ratio value was as follows:”.
Response:Thanks. I have updated it.
Lines 146, 150, and 155: “where” or “Where”? Be consistent.
Response:Thanks. I have updated it.
Lines 163 and 165: “using” instead of “by”.
Response:Thanks. I have updated it.
Line 178: Delete “1”.
Response:Thanks. I have updated this sentence.
Line 179: Do not begin a sentence with “And”. Change “5” to “five”.
Response:Thanks. I have updated it.
Line 183: It does not make sense to say the accuracy is more accurate. Reword.
Response:Thanks. I have rewritten this sentence. Hope that it looks better.
Line 185: Add a comma in “cov (a p)”.
Response:Thanks. I have updated it.
Line 191: “are shown” rather than “was shown”. I do not see Table 1.
Response:Thanks. I have updated it. In the revised version, Hope that you can see all the tables.
Lines 192-193: Change to “…and 37.20 kg, respectively, and the coefficients of variation ranged from…”.
Response:Thanks. I have updated it.
Line 197: I do not see Table 2.
Response:Hope that you can see all the tables.
Lines 198-205: P < 0.05 indicates significant differences between means. P < 0.01 shows highly significant differences between means. P < 0.001 does not have a description such as extremely significant.
Response:In the revised version, these were shown in Note for table1.
Line 208: Where is Table 3?
Line 221: I cannot find Tables 5 – 8.
Response:Sorry! Perhaps you have reviewed an incorrect version, the final version we submitted included the table. I hope this version is correct, as you can see the tables.
Lines 224, 229, 233, and 238: “interaction” or “correlation”?
Response:Thanks. I have updated this.
Line 247: I cannot find Table 9.
Response:Sorry! Perhaps you have reviewed an incorrect version, the final version we submitted included the table. I hope this version is correct, as you can see the tables.
Line 251: Do not begin a sentence with “And”.
Response:Yes, I have updated it.
Line 256: Delete “so”.
Response:Yes, I have deleted it.
Line 257: Insert “and” before “herds”.
Response:Yes, I have updated it.
Line 260: Delete “extremely or”.
Response:Yes, I have deleted it.
Lines 271-273: This is an incomplete sentence and must be rewritten.
Response:I have updated this sentence. Hope that it looks better.
Line 273: Delete “It was obtained that”.
Response:Yes, I have deleted it.
Lines 276-277: Why would the reproductive ability of dams influence early growth rate? What do you mean by ewes’ body function? Be more specific.
Response:I have updated this sentence. Hope that it looks better.
Line 277: “within” does not need to be capitalized.
Response:Thanks. I have deleted it.
Lines 283-285: Here is a clearer and more grammatically correct version of your sentence:
"In this study, four animal models were constructed to assess the maternal genetic effect, maternal environmental effect, and the covariance between the direct additive effect and maternal genetic effect."
Response:Thanks. I have updated it.
Line 287: Change “interaction” to “covariance”.
Response:Thanks. I have updated it.
Lines 291-292: This is an incomplete sentence. Reword.
Response:Thanks. I have rewritten it.
Line 294: “Black” should be capitalized since it is part of the breed’s name. “is” instead of “was”.
Response:Thanks. I have updated it.
Line 295: Change to “…genetic parameters. The results…”.
Response:Thanks. I have updated it.
Line 307: “Merino” needs to be capitalized since it is the name of a breed.
Response:Thanks. I have updated it.
Line 308: “Merinos” must be capitalized since it is part of the breed’s name.
Response:Thanks. I have updated it.
Lines 308-309: “All of which were similar to the results in our study” is an incomplete sentence. Reword.
Response:I have updated this sentence.
Line 315: “Angora goats Black goats” does not make sense. What is the correct name of the breed?
Response:Thanks for your reminder. I have updated this.
Line 316: Leave a space between the sentences. What is “It”?
Response:I have updated this sentence. Hope that it looks better.
Line 321: What is “descent determination”?
Response: it means “progeny testing”. I have updated it. Thanks for your suggestion.
Line 327: Change to “…was almost as high as…”.
Response:Thanks. I have updated it.
Line 329: Insert “and” before “Bayes”.
Response:Thanks. I have updated it.
Lines 341-343: Change to “Animal models for genetic evaluation of early growth traits in IMCGs should include direct additive genetic effects, maternal genetic effects, maternal environmental effects and the covariance between direct additive and maternal genetic effects”.
Response:Thanks. I have updated it.
Do the maternal permanent environmental effects need to be included in the model given that they accounted for a tiny proportion of the phenotypic variance in your study?
Response: I agree with your opinion. It is a very good view. Our team recently conducted a genetic evaluation of early growth traits using another dataset with six models. Indeed, the genetic evaluation of early growth traits can disregard the effects of the maternal permanent environment. In this study, Model 3 and Model 4 reached a significant level through likelihood ratio testing. Adding maternal permanent environmental effects to the model will not affect the results of genetic evaluation for early growth traits, including the estimation of genetic parameters and breeding values.
Line 345: Here and throughout the manuscript, change “daily gain weight” to “daily weight gain”.
Response:I have updated this in the whole text.
Lines 360-364: Only use the authors’ initials in this section.
Response:Thanks. I have updated it.

Round 2
Reviewer 2 Report
Comments and Suggestions for Authors
The authors have thoroughly reviewed the aspects highlighted and therefore, I consider that the work does not require further revisions.
Author Response
Thank you very much for your comments.
Reviewer 3 Report
Comments and Suggestions for Authors
The manuscript requires editing for clarity and conciseness.
Line 20: Who is “they”? Do you mean “we”?
Line 22: “dam” instead of “maternal”.
Line 29: Delete “(1) Background:”.
Lines 30-31: Delete “(2) Methods:”.
Line 33: Insert “a” before “generalized”.
Line 36: Insert “a” before “Likelihood”. Likelihood does not need to be capitalized. Remove the space before the period at the end of the sentence.
Lines 37-38: Delete “(3) Results:”.
Line 43: Correct the spelling of “under”.
Line 44: Insert “were” before “0.11”.
Line 49: You have not been capitalizing the word “cashmere”. Be consistent.
Line 56-57 and 63: “performance” instead of “performances”.
Lines 62, 65, and 67: Leave a space before the left bracket here and throughout the manuscript.
Lines 74-75: “had been performed genomic selection” is unclear. Reword.
Line 88: “traits” instead of “trait”.
Line 94: “dairy weight”?
Lines 93-94: Delete “The traits including”.
Line 95: Insert “and” before “yearling”. “phenotypes” instead of “phenotype”.
Line 97: Delete the space in “70 K”.
Line 101: “genotypes” instead of “genotype”.
Lines 106-107: Leave space after the right parenthesis.
Line 111: Change the subheading to “Determination of fixed and random effects”.
Line 112: Change to “is essential”.
Line 114: Delete the space before the commas.
Line 115: What is a “raw herd”?
Line 138: Change the periods to commas.
Line 158: “matrix” instead of “matrices”.
Lines 160 and 165: Correct the spelling of “inverse”.
Line 180: Do not begin a sentence with “And”. Delete “And”.
Line 185: “follows” instead of “follow”.
Line 194: Change to “…model equations. It reflects the…”.
Lines 206-207: Leave space after the colons.
Line 207: Leave a space before “Max”.
Line 213: Change to “… four traits was highly significant (P < 0.001)”.
Lines 213-216: Change to “Year of birth had a highly significant effect on DWG and WW (P < 0.001), and on WW (P < 0.01). Herd had a highly significant impact on WW and DWG (P < 0.01), and on YW (P < 0.001)”. Note that WW is listed twice in the first sentence. There is no such statistical terminology as “extremely significant”.
Lines 216-210: Change to “Age of dam had a highly significant effect on all the traits in this study (P < 0.01). The impact of birth type was highly significant for BW, WW, and DWG (P < 0.001) and YW (P < 0.01). Therefore, all the factors were included as fixed effects in the animal models used for genetic evaluation in this study.”
Line 221: Leave space after the colons.
Line 222: Change “extremely” to “highly”. Extremely significant is not proper statistical terminology.
Line 226: Delete “It is known that”.
Line 227: Change “is” to “are”.
Lines 227-229: Change to “…each method are lower than that in the other models, indicating that model 4 is the best model for estimating genomic breeding values of early growth traits of IMCGs”.
Line 231: Delete “the”. Change “difference” to “differences”.
Line 235: Insert “the” before “ABLUP”.
Line 237: Leave a space before “in”.
Tables 3 and 4: Define the trait abbreviations in a footnote.
Table 4: The column heading “Model” must be all on one line.
Lines 245-246: Do you mean “optimal model” instead of “optional model”?
Lines 248-249: I do not understand “2.45 × 10-8 -0.12”. Please explain or modify as needed.
Lines 250-251: Change to “…genetic effects was negative in the ABLUP and GBLUP analyses. The correlation coefficients were -0.41 and -0.35, respectively (Table 5)”.
Line 252: Insert “the” before “three”.
Line 253: Change the semicolon to a period.
Line 254: Change to “… was 1.90 × 10-8 to -1.36 × 10-5.”
Lines 255-256: Change to “…effects was negative. The correlation coefficients varied from -0.54 to -0.91 (Table 6).
Line 257: Insert “the” before “three”.
Lines 258-261: Change to “was 5.84 × 10-3 to -7.49 × 10-3. The covariance between direct and maternal additive genetic effects was negative. The correlation coefficients ranged from -0.41 to -0.32 (Table 7)”.
Line 261: Insert “the” before “three”.
Lines 264-266: Change to “The covariance between direct additive and maternal additive genetic effects was negative. The correlation coefficients ranged from -0.84 to -0.56 (Table 8).
Line 272: “birth” should not be capitalized.
Lines 274-278: Use consistent spacing before and after the colons. Leave space after the semicolons.
Line 280: “weaning” should not be capitalized.
Lines 282-286: Use consistent spacing before and after the colons. Leave space after the semicolons.
Lines 291-295: Use consistent spacing before and after the colons. Leave space after the semicolons.
Line 298: “yearling” should not be capitalized.
Lines 300-304: Use consistent spacing before and after the colons. Leave space after the semicolons.
Tables 5-8: Leave space before and after the ± sign in the column headings.
Line 309: Delete the space after the hyphen in “0.68- 0.70”.
Line 311: Insert “the” before “ssGBLUP”.
Line 312: Insert “the” before “other”.
Table 9: Define superscripts a, b, and c in a footnote.
Line 335: Change to “…on the farms”.
Lines 338-339: By “physiological status of ewes,” do you mean milk production?
Lines 343-344: It is obvious that “early growth traits have important effects on growth and development of animals”. This statement should be deleted.
Lines 347-350: Listing the four effects again is unnecessary. Say, “The model including all four effects best estimated the genetic parameters of the early growth traits in IMCGs”.
Line 352: Replace “effect” with “effects”.
Lines 354-356: This is an incomplete sentence. Since this is the case, then what?
Line 354: What is “pregnancy ability”? Conception rate?
Line 357: Change to “are ignored”.
Line 359: Change to “…parameters for body weight…”.
Line 367: “pre-weaning (100-day)”? Weight? Daily weight gain? Milk production? Please specify the trait.
Line 384: Delete “by”.
Line 390: Change to “…highest using ssGBLUP”.
Line 391: Delete “method”.
Line 392: Change to “…the GBLUP method was almost as high as that with the Bayes…”.
Line 395: Change to “…than the other methods”.
Lines 400-401: Delete “method” in both places.
Line 401: “genomic” instead of “genome”.
Line 409: Insert “an” before “impact”.
Line 410: Change to “…were 0.09 - 0.11 and 0.12 - 0.15, respectively”.
Line 411: Change to “Both were lowly heritable traits”. Heritabilities of 0.15 or less are low, not moderate.
Lines 412-413: Change to “The direct heritability estimates for WW and YW were moderate (0.17 - 0.43 and 0.20 - 0.32, respectively)”. Heritabilities of 0.43 or less are not high.
Line 418: Insert “the” before “ssGBLUP”.
Lines 426-430: Use only the author’s initials in this section.
Line 437: Leave a space after the colon.
Line 440: “were” in place of “was”. The word “data” is plural.
Line 443: Do not use contractions in scientific writing. Change “didn’t” to “did not”.
Comments on the Quality of English Language
The manuscript requires a great deal of editing for clarity, conciseness, grammar, and format.
Author Response
Dear reviewer,
Thank you very much for your extensive language editing for my manuscript, which has significantly improved its quality. Additionally, I have also sought assistance from the MDPI language editing team to refine it further. I hope this version is acceptable. Thanks.
Line 20: Who is “they”? Do you mean “we”?
Answer: yes. It means “we”.
Line 22: “dam” instead of “maternal”.
Answer: Thanks for your advice. I have revised it.
Line 29: Delete “(1) Background:”.
Answer: I have revised it. Thanks.
Lines 30-31: Delete “(2) Methods:”.
Answer: I have revised it. Thanks.
Line 33: Insert “a” before “generalized”.
Answer: I have revised it. Thanks.
Line 36: Insert “a” before “Likelihood”. Likelihood does not need to be capitalized. Remove the space before the period at the end of the sentence.
Answer: I have revised it. Thanks.
Lines 37-38: Delete “(3) Results:”.
Answer: I have revised it. Thanks.
Line 43: Correct the spelling of “under”.
Answer: I have revised it. Thanks.
Line 44: Insert “were” before “0.11”.
Answer: I have revised it. Thanks.
Line 49: You have not been capitalizing the word “cashmere”. Be consistent.
Answer: I have revised it. Thanks.
Line 56-57 and 63: “performance” instead of “performances”.
Answer: I have revised it. Thanks.
Lines 62, 65, and 67: Leave a space before the left bracket here and throughout the manuscript.
Answer: I have revised it. Thanks.
Lines 74-75: “had been performed genomic selection” is unclear. Reword.
Answer: I have updated this sentence. Hope that it is OK.
Line 88: “traits” instead of “trait”.
Answer: I have revised it. Thanks.
Line 94: “dairy weight”?
Answer: I have updated it. Thanks.
Lines 93-94: Delete “The traits including”.
Answer: I have revised it. Thanks.
Line 95: Insert “and” before “yearling”. “phenotypes” instead of “phenotype”.
Answer: I have revised it. Thanks.
Line 97: Delete the space in “70 K”.
Answer: I have revised it. Thanks.
Line 101: “genotypes” instead of “genotype”.
Answer: I have revised it. Thanks.
Lines 106-107: Leave space after the right parenthesis.
Answer: I have revised it. Thanks.
Line 111: Change the subheading to “Determination of fixed and random effects”.
Answer: I have revised it. Thanks.
Line 112: Change to “is essential”.
Answer: I have revised it. Thanks.
Line 114: Delete the space before the commas.
Answer: I have revised it. Thanks.
Line 115: What is a “raw herd”?
Answer: Sorry. It is a spelling error.
Line 138: Change the periods to commas.
Answer: I have revised it. Thanks.
Line 158: “matrix” instead of “matrices”.
Answer: I have revised it. Thanks.
Lines 160 and 165: Correct the spelling of “inverse”.
Answer: I have revised it. Thanks.
Line 180: Do not begin a sentence with “And”. Delete “And”.
Answer: I have revised it. Thanks.
Line 185: “follows” instead of “follow”.
Answer: I have revised it. Thanks.
Line 194: Change to “…model equations. It reflects the…”.
Answer: I have revised it. Thanks.
Lines 206-207: Leave space after the colons.
Answer: I have revised it. Thanks.
Line 207: Leave a space before “Max”.
Answer: I have revised it. Thanks.
Line 213: Change to “… four traits was highly significant (P < 0.001)”.
Answer: I have revised it. Thanks.
Lines 213-216: Change to “Year of birth had a highly significant effect on DWG and WW (P < 0.001), and on WW (P < 0.01). Herd had a highly significant impact on WW and DWG (P < 0.01), and on YW (P < 0.001)”. Note that WW is listed twice in the first sentence. There is no such statistical terminology as “extremely significant”.
Answer: I have revised it. Thanks.
Lines 216-210: Change to “Age of dam had a highly significant effect on all the traits in this study (P < 0.01). The impact of birth type was highly significant for BW, WW, and DWG (P < 0.001) and YW (P < 0.01). Therefore, all the factors were included as fixed effects in the animal models used for genetic evaluation in this study.”
Answer: I have revised it. Thanks.
Line 221: Leave space after the colons.
Answer: I have revised it. Thanks.
Line 222: Change “extremely” to “highly”. Extremely significant is not proper statistical terminology.
Answer: I have revised it. Thanks.
Line 226: Delete “It is known that”.
Answer: I have revised it. Thanks.
Line 227: Change “is” to “are”.
Answer: I have revised it. Thanks.
Lines 227-229: Change to “…each method are lower than that in the other models, indicating that model 4 is the best model for estimating genomic breeding values of early growth traits of IMCGs”.
Answer: I have revised it. Thanks.
Line 231: Delete “the”. Change “difference” to “differences”.
Answer: I have revised it. Thanks.
Line 235: Insert “the” before “ABLUP”.
Answer: I have revised it. Thanks.
Line 237: Leave a space before “in”.
Answer: I have revised it. Thanks.
Tables 3 and 4: Define the trait abbreviations in a footnote.
Answer: I have revised it. Thanks.
Table 4: The column heading “Model” must be all on one line.
Answer: I have revised it. Thanks.
Lines 245-246: Do you mean “optimal model” instead of “optional model”?
Answer: I have revised it. Thanks.
Lines 248-249: I do not understand “2.45 × 10-8 -0.12”. Please explain or modify as needed.
Answer: Sorry, it may be related to methods. I don not know how to modify it.
Lines 250-251: Change to “…genetic effects was negative in the ABLUP and GBLUP analyses. The correlation coefficients were -0.41 and -0.35, respectively (Table 5)”.
Answer: I have revised it. Thanks.
Line 252: Insert “the” before “three”.
Answer: I have revised it. Thanks.
Line 253: Change the semicolon to a period.
Answer: I have revised it. Thanks.
Line 254: Change to “… was 1.90 × 10-8 to -1.36 × 10-5.”
Answer: I have revised it. Thanks.
Lines 255-256: Change to “…effects was negative. The correlation coefficients varied from -0.54 to -0.91 (Table 6).
Answer: I have revised it. Thanks.
Line 257: Insert “the” before “three”.
Answer: I have revised it. Thanks.
Lines 258-261: Change to “was 5.84 × 10-3 to -7.49 × 10-3. The covariance between direct and maternal additive genetic effects was negative. The correlation coefficients ranged from -0.41 to -0.32 (Table 7)”.
Answer: I have revised it. Thanks.
Line 261: Insert “the” before “three”.
Answer: I have revised it. Thanks.
Lines 264-266: Change to “The covariance between direct additive and maternal additive genetic effects was negative. The correlation coefficients ranged from -0.84 to -0.56 (Table 8).
Answer: I have revised it. Thanks.
Line 272: “birth” should not be capitalized.
Answer: I have revised it. Thanks.
Lines 274-278: Use consistent spacing before and after the colons. Leave space after the semicolons.
Answer: I have revised it. Thanks.
Line 280: “weaning” should not be capitalized.
Answer: I have revised it. Thanks.
Lines 282-286: Use consistent spacing before and after the colons. Leave space after the semicolons.
Answer: I have revised it. Thanks.
Lines 291-295: Use consistent spacing before and after the colons. Leave space after the semicolons.
Answer: I have revised it. Thanks.
Line 298: “yearling” should not be capitalized.
Answer: I have revised it. Thanks.
Lines 300-304: Use consistent spacing before and after the colons. Leave space after the semicolons.
Answer: I have revised it. Thanks.
Tables 5-8: Leave space before and after the ± sign in the column headings.
Answer: I have revised it. Thanks.
Line 309: Delete the space after the hyphen in “0.68- 0.70”.
Answer: I have revised it. Thanks.
Line 311: Insert “the” before “ssGBLUP”.
Answer: I have revised it. Thanks.
Line 312: Insert “the” before “other”.
Answer: I have revised it. Thanks.
Table 9: Define superscripts a, b, and c in a footnote.
Answer: I have revised it. Thanks.
Line 335: Change to “…on the farms”.
Answer: I have revised it. Thanks.
Lines 338-339: By “physiological status of ewes,” do you mean milk production?
Answer: No, it also include other factors.
Lines 343-344: It is obvious that “early growth traits have important effects on growth and development of animals”. This statement should be deleted.
Answer: I have revised it. Thanks.
Lines 347-350: Listing the four effects again is unnecessary. Say, “The model including all four effects best estimated the genetic parameters of the early growth traits in IMCGs”.
Answer: I have revised it. Thanks.
Line 352: Replace “effect” with “effects”.
Answer: I have revised it. Thanks.
Lines 354-356: This is an incomplete sentence. Since this is the case, then what?
Answer: I have revised it. Thanks.
Line 354: What is “pregnancy ability”? Conception rate?
Answer: Yes, it is equal.
Line 357: Change to “are ignored”.
Answer: I have revised it. Thanks.
Line 359: Change to “…parameters for body weight…”.
Answer: I have revised it. Thanks.
Line 367: “pre-weaning (100-day)”? Weight? Daily weight gain? Milk production? Please specify the trait.
Answer: I have revised it. Thanks.
Line 384: Delete “by”.
Answer: I have revised it. Thanks.
Line 390: Change to “…highest using ssGBLUP”.
Answer: I have revised it. Thanks.
Line 391: Delete “method”.
Answer: I have revised it. Thanks.
Line 392: Change to “…the GBLUP method was almost as high as that with the Bayes…”.
Answer: I have revised it. Thanks.
Line 395: Change to “…than the other methods”.
Answer: I have revised it. Thanks.
Lines 400-401: Delete “method” in both places.
Answer: I have revised it. Thanks.
Line 401: “genomic” instead of “genome”.
Answer: I have revised it. Thanks.
Line 409: Insert “an” before “impact”.
Answer: I have revised it. Thanks.
Line 410: Change to “…were 0.09 - 0.11 and 0.12 - 0.15, respectively”.
Answer: I have revised it. Thanks.
Line 411: Change to “Both were lowly heritable traits”. Heritabilities of 0.15 or less are low, not moderate.
Answer: I have revised it. Thanks.
Lines 412-413: Change to “The direct heritability estimates for WW and YW were moderate (0.17 - 0.43 and 0.20 - 0.32, respectively)”. Heritabilities of 0.43 or less are not high.
Answer: I have revised it. Thanks.
Line 418: Insert “the” before “ssGBLUP”.
Answer: I have revised it. Thanks.
Lines 426-430: Use only the author’s initials in this section.
Answer: I have revised it. Thanks.
Line 437: Leave a space after the colon.
Answer: I have revised it. Thanks.
Line 440: “were” in place of “was”. The word “data” is plural.
Answer: I have revised it. Thanks.
Line 443: Do not use contractions in scientific writing. Change “didn’t” to “did not”.
Answer: I have revised it. Thanks.